

**Distribution of planktonic biogenic carbonate organisms in the Southern Ocean south of**
**Australia: a baseline for ocean acidification impact assessment**
Thomas W. Trull[1,2,3], Abraham Passmore[1,2], Diana M. Davies[1,2], Tim Smit[4], Kate Berry[1,2], and Bronte
Tilbrook[1,2]
1. Climate Science Centre, Oceans and Atmosphere, Commonwealth Scientific and Industrial
Research Organisation, Hobart, 7001, Australia
2. Antarctic Climate and Ecosystems Cooperative Research Centre, Hobart, 7001, Australia
3. Institute of Marine and Antarctic Studies, University of Tasmania, Hobart, 7001, Australia
4. Utrecht University, Utrecht, 3508, Holland
*Correspondence to*: Tom Trull (Tom.Trull@csiro.au)

15                                      **Abstract**

The Southern Ocean provides a vital service by absorbing about one sixth of humankind's annual
emissions of $CO_2$. This comes with a cost – an increase in ocean acidity that is expected to have
negative impacts on ocean ecosystems. The reduced ability of phytoplankton and zooplankton to
precipitate carbonate shells is a clearly identified risk. The impact depends on the significance of these
organisms in Southern Ocean ecosystems, but there is very little information on their abundance or
distribution.  To quantify their presence, we used coulometric measurement of particulate inorganic
carbonate (PIC) on particles filtered from surface seawater into two size fractions: 50-1000 μm to
capture foraminifera (the most important biogenic carbonate forming zooplankton) and 1-50 μm to
capture coccolithophores (the most important biogenic carbonate forming phytoplankton).  Ancillary
measurements of biogenic silica (BSi) and particulate organic carbon (POC) provided context, as
estimates of the abundance of diatoms (the most abundant phytoplankton in polar waters), and total
microbial biomass, respectively.  Results for 9 transects from Australia to Antarctica in 2008-2015
showed low levels of PIC compared to northern hemisphere polar waters. Coccolithophores slightly
exceeded the biomass of diatoms in Subantarctic waters, but their abundance decreased more than 30-
fold poleward, while diatom abundances increased, so that on a molar basis PIC was only 1% of BSi
in Antarctic waters.  This limited importance of coccolithophores in the Southern Ocean is further
emphasized in terms of their associated POC, representing less than 1 % of total POC in Antarctic
waters and less than 10% in Subantarctic waters.   NASA satellite ocean colour based PIC estimates
were in reasonable agreement with (though somewhat higher than) the shipboard results in
Subantarctic waters, but greatly over-estimated PIC in Antarctic waters.  Contrastingly, the NASA
Ocean Biogeochemical Model (NOBM) shows coccolithophores as overly restricted to Subtropical



and northern Subantarctic waters.  The cause of the strong southward decrease in PIC abundance in
the Southern Ocean is not yet clear.  Poleward decrease in pH is small and while calcite saturation
decreases strongly southward it remains well above saturation (>2).  Nitrate and phosphate variations
would predict a poleward increase.  Temperature and competition with diatoms for limiting iron
appear likely to be important. While the future trajectory of coccolithophore distributions remains
uncertain, their current low abundances suggest small impacts on overall Southern Ocean pelagic
ecology.



## 1. Introduction

Production of carbonate minerals by planktonic organisms is an important and complex part of the global carbon cycle and climate system. On the one hand, carbonate precipitation raises the partial pressure of $CO_2$ reducing the uptake of carbon dioxide from the atmosphere into the surface ocean; on the other hand the high density and slow dissolution of these minerals promotes the sinking of associated organic carbon more deeply into the ocean interior increasing sequestration [*P.W. Boyd and Trull*, 2007b; *Buitenhuis et al.*, 2001; *Klaas and Archer*, 2002; *Ridgwell et al.*, 2009; *Salter et al.*, 2014]. Carbonate production is expected to be reduced by ocean acidification from the uptake of anthropogenic $CO_2$, with potentially large consequences for the global carbon cycle and ocean ecosystems [*Orr et al.*, 2005; *Pörtner et al.*, 2005].

The naturally low alkalinity of Southern Ocean waters makes this region particularly susceptible to ocean acidification impacts, in that thresholds such as undersaturation of aragonite and calcite carbonate minerals will be crossed sooner in this region than at lower latitudes [*Cao and Caldeira*, 2008; *McNeil and Matear*, 2008; *Shadwick et al.*, 2013]. Important planktonic organisms include coccolithophores (the dominant carbonate forming phytoplankton; e.g. [*Rost and Riebesell*, 2004]), foraminifera (the dominant carbonate forming zooplankton; e.g. [*Moy et al.*, 2009; *Schiebel*, 2002]), and pteropods (a larger carbonate forming zooplankton, which can be an important component of fish diets; e.g. [*Doubleday and Hopcroft*, 2015; *Roberts et al.*, 2014]). The importance of carbonate forming organisms relative to other taxa, which is poorly known in the Southern Ocean [*Watson W. Gregg and Casey*, 2007b; *Holligan et al.*, 2010], will influence the overall impact of ocean acidification on ecosystem health. Satellite reflectance observations, mainly calibrated against northern hemisphere PIC results, have been interpreted to suggest the presence of a "Great Calcite Belt" in Subantarctic waters in the Southern Ocean , and also show high apparent PIC values in Antarctic waters [*W M Balch et al.*, 2016; *W M Balch et al.*, 2011]. Our surveys were designed in part to evaluate these assertions for waters south of Australia.

As a simple step towards quantifying the importance of planktonic biogenic carbonate forming organisms in the Southern Ocean, we determined the concentrations of particulate inorganic carbonate (PIC) for two size classes, representing coccolithophores (1-50 □m, referred to as PIC01) and foraminifera (50-1000 μm, referred to as PIC50), from surface water samples collected on 9 transects between Australia and Antarctica. We provide ecological context for these observations based on the abundance of particulate organic carbon (POC) as a measure of total microbial biomass, and biogenic silica (BSi), the other major phytoplankton biogenic mineral, as a measure of diatom biomass. This provides a baseline assessment of the importance of calcifying plankton in the Southern Ocean south



of Australia, against which future levels can be compared.  The baseline suggests lower PIC
abundances that suggested from the current satellite SPIC algorithms, especially in Antarctic waters.

In the discussion of our results, we interpret the BSi results as representative of diatoms, the PIC50 as
representative of foraminifera, and the PIC01 as representative of coccolithophores, including a
tendency to equate this with the distribution of the most cosmopolitan and best studied
coccolithophore, *Emiliania huxleyi*.  These assumptions need considerable qualification.  Most BSi is
generated by diatoms (~90%), with only minor contributions from radiolaria and choanoflagellates in
the upper ocean, making this approximation reasonably well supported [*Hood et al.*, 2006].  Similarly,
but less certainly, foraminifera are a major biogenic carbonate source in the 50-1000 μm size range,
but pteropods, ostrocods, and other organisms are also important [*Schiebel*, 2002], so that this
approximation is weaker.  We do not discuss the PIC50 results in any detail partly for this reason, but
more importantly because controls on foraminifera distributions appear to involve strongly differing
biogeography of several co-dominant taxa, rather than dominance by a single species [*Be and*
*Tolderlund*, 1971], and assessing these issues is beyond the scope of this paper. Attributing all the
PIC01 carbonate to coccolithophores relies on the assumption that fragments of larger organisms are
not important.  This seems reasonable given that the larger PIC50 fraction generally contained 10-fold
lower PIC concentrations (as revealed in the Results section).

Our tendency to equate the PIC01 fraction with the abundance of *Emiliania huxleyi* is probably the
weakest approximation.  It is not actually central to our conclusions, except to the extent that we
compare our PIC01 distributions to expectations based on models that use physiological results
mainly derived from experiments with this species. That said, this is a poor approximation in
Subtropical waters where the diversity of coccolithophores is large, but improves southward where
the diversity decreases (see Smith et al. 2017 for recent discussion), and many observations have
found that *Emiliania huxleyi* was strongly dominant in Subantarctic and Antarctic Southern Ocean
populations, generally >80% [*Boeckel et al.*, 2006; *Eynaud et al.*, 1999; *Findlay and Giraudeau*,
2000; *Gravalosa et al.*, 2008; *Mohan et al.*, 2008]. Of course, *Emiliania huxleyi* itself comes in
several strains even in the Southern Ocean, with differing physiology [*Cubillos et al.*, 2007; *M. N.*
*Muller et al.*, 2015; *M.N. Muller et al.*, 2017]. All these approximations are important to keep in mind
in any generalization of our results.

**2. Methods**
Sub-sections 2.1 and 2.2 present the sampling and analytical methods, respectively, used for the 8
transits across the Southern Ocean since 2012.  Sub-section 2.3 details the different methods used
during the earlier single transit in 2008 and assesses the comparability of those results to the later
voyages. Sub-section 2.4 details measurements of water column dissolved nutrients, inorganic carbon





and alkalinity.  Sub-section 2.5 provides details of satellite remote sensing data and the NASA Ocean
Biogeochemical Model used for comparison to the ship results.

**2.1. Voyages and sample collection procedures**
The locations of the voyages, divided into north and south legs, are shown in Figure 1. Voyage and
sample collection details are given in Table 1, where for ease of reference we have numbered the legs
in chronological order and refer to them hereafter as VL1, VL2, etc. Samples were collected from the
Australian icebreaker *RV Aurora Australia* for 4 voyages and from the French Antarctic resupply
vessel *l'Astrolabe* for 1 voyage.  All samples were collected from the ships' clean seawater supplies
with intakes at ~4 m depth.  Samples were collected primarily while underway, except during VL1
and VL3, which were operated as WOCE/CLIVAR hydrographic sections with full depth CTDs, with
samples collected on station.

For all voyages (except VL1, discussed in section 2.3 below), separate water volumes were collected
for the PIC, POC, and BSi analyses.  The POC samples also yielded particulate nitrogen results -
referred to here as PON.  The POC/PON and BSi samples were collected using a semi-automated
system that rapidly, ~ 1 minute, and precisely filled separate 1 L volumes for each analyte - thus these
samples are effectively point samples.  In contrast, PIC samples were collected using the pressure of
the underway seawater supply to achieve filtration of large volumes (10's to 100's of litres) over ~2
hours. Thus these samples represent collections along ~20 miles of the ship track (except when done
at stations).

POC/PON samples were filtered through pre-combusted 13 mm diameter quartz filters (0.8 µm pore
size, Sartorius Cat#FT-3-1109-013) that had been pre-loaded in clean (flow-bench) conditions in the
laboratory into in-line polycarbonate filter holders (Sartorius #16514E).  The filters were preserved by
drying in their filter holders at 60°C for 48 hours at sea, and returned to the laboratory in clean dry
boxes.

Biogenic silica samples were filtered through either 13 mm diameter nitrocellulose filters (0.8 µm
pore size, Millipore Cat#AAWP01300) or 13 mm diameter polycarbonate filters (0.8 µm pore size,
Whatman Cat#110409), pre-loaded in clean (flow-bench) conditions in the laboratory into in-line
polycarbonate filter holders (Sartorius #16514E). Filters were preserved by drying in their filter
holders at 60°C for 48 hours at sea, and returned to the laboratory in clean dry boxes.

PIC samples were collected by sequential filtration for two size fractions. After pre-filtration through
a 47 mm diameter 1000 µm nylon mesh and supply pressure reduction to 137 kPa, the ship clean
seawater was filtered  through a 47 mm diameter in-line 50 µm nylon filter to collect foraminifera,



and then through a 47 mm diameter in-line 0.8 μm GF/F filter (Whatman Cat#1825-047) to collect
coccolithophores.  The flow path was split using a pressure relief valve set to 55 kPa, so that large
volumes (~200 L) passed the 50 μm filter, and only a small fraction of this volume (~15 L) passed the
0.8 μm filter. Filtration time was typically 2 hours. Volume measurement was done by either metering
or accumulation. While still in their holders, the filters were rinsed twice with 3 mL of 20 mM
potassium tetraborate buffer solution (for the first couple of voyages and later deionized water) to
remove dissolved inorganic carbon, and blown dry with clean pressurised air (69 kPa), The filters
were then removed from their holders, folded, and inserted into Exetainer glass tubes (Labco Cat
#938W) and dried at 60 °C for 48 hours for return to the laboratory.  In the following text, we refer to
the GF/F filter sample results (which sampled the 0.8 (~ 1) to 50 μm size fraction) as PIC01, and the
nylon mesh sample fraction (which sampled the 50-1000 μm size fraction) as PIC50.

**2.2 Sample analyses**
**2.2.1 Particulate Organic Carbon and Nitrogen analysis**
The returned filter holders were opened in a laminar flow bench and the filters cleanly transferred into
silver cups (Sercon Cat#SC0037), acidified with 50 μL of 2 N HCl and incubated at room temperature
for 30 minutes to remove carbonates, and dried in an oven at 60 °C for 48 hours. The silver cups were
then folded closed and the samples, along with process blanks (filters treated in the same way as
samples, but without any water flow onboard the ship) and casein standards (Elemental Microanalysis
OAS standard CatNo. B2155, Batch 114859) were sent to the University of Tasmania Central
Sciences Laboratory for CHN elemental analysis against sulphanilamide standards. Precision of these
analyses, based on standard variations was a few percent for POC and PON, but importantly the
processing blanks were larger and variable, and were corrected for separately for each voyage.  For
VL2 and VL3, POC processing blanks averaged 25± 6 μg C (1 sd, n=2) equating to 20% of average
sample values. For VL4 and VL5, POC process blanks averaged $14 \pm 2$ μg C (1 sd, n=4) equating to
18% of average sample values. For VL6 and Vl7, POC process blanks averaged $23 \pm 3$ μg C (1 sd
n=4) equating to 28 % of average sample values.

**2.2.2 Biogenic Silica analysis**
Biogenic silica was dissolved by adding 4 mL of 0.2 M NaOH and incubating at 95 °C for 90 minutes,
similar to the method of [*Paasche*, 1973]. Samples were then rapidly cooled to 4 °C and neutralized
with 1 mL of 1 M HCl. Thereafter samples were centrifuged at 1880 *g* for 10 minutes and the
supernatant was transferred to a new tube and diluted with artificial seawater (36 g L$^{-1}$ NaCl).
Biogenic silica concentrations were determined by spectrophotometry using an Alpkem model 3590
segmented flow analyser and following USGS Method I-2700-85 with these modifications:
ammonium molybdate solution contained 10 g L$^{-1}$ (NH$_4$)$_6$Mo$_7$O$_{24}$, 800 μl of 10% sodium dodecyl



sulphate detergent replaced Levor IV solution, acetone was omitted from the ascorbic acid solution,
and artificial seawater was used as the carrier solution.

Biogenic silica standard concentrations were 0, 28, 56, 84, 112 and 140 µM. The sensitivity of
standard curves (forced through 0) varied by less than 1% (1 sd, n=5). The mean concentration of
repeated check standards (140 µM) run after every 12 samples was 140±0.5 µM (1 sd, n=64). The
average blank value was $0.014 \pm 0.003$ µmoles per filter (1 sd, n=9) for nitrocellulose filters and 0.010
$\pm 0.005$ µmoles per filter (1 sd, n=6) for polycarbonate filters, equating to ~2 % and 1.5 % of average
sample values, respectively.

**2.2.3 Particulate Inorganic Carbon analysis**
Particulate inorganic carbon samples were analysed by coulometry using a UIC CM5015 coulometer
connected to a Gilson 232 autosampler. The samples were analysed directly in their collection tubes,
by purging for 5 minutes with nitrogen gas, acidification with 1.6 mL (PIC50 - 50 µm nylon filters) or
2.4 mL (PIC01 - GF/F filters) of 1 N phosphoric acid, and equilibration overnight at 40°C. Samples
were analysed the following day with a sample analysis time of 8 minutes and a dried carrier gas flow
rate of 160 mL min$^{-1}$. Calcium carbonate standards (Sigma Cat#398101-100G) were either weighed
onto GF/F filters or weighed into tin cups (Sercon Cat# SC1190) and then inserted into Exetainer
tubes (some with blank nylon filters). Typical standard weights were circa 0, 50, 200, 1500 and 6500
µg. Standard curves for GF/F filters (forced through 0) across all voyages varied by less than 0.9% (1
sd, n=9), and for nylon filters by less than 0.6% (1 sd, n=10). The mean percentage recovery of
repeated check standards for GF/F filters was $100.5 \pm 3.9$ % (1 sd, n=29), and for nylon mesh filters
$100.2 \pm 1.9$ % (1 sd n=30). The average GF/F filter blank value was $0.67 \pm 0.26$ µg C (1 sd, n=15)
equating to 2% of average sample values, and for nylon filters was $0.56 \pm 0.19$ µg C (1 sd, n=21)
equating to 0.9% of average sample values.

**2.3 Distinct sample collection and analytical methods used during V1**
**2.3.1 Distinct sample collection procedures for VL1**
For VL1, single samples were collected at each location by both sequential filtration and
centrifugation of the underway supply over 1-3 hours.  Despite the long collection times these
samples are effectively point samples because they were collected on station.

Sequential filtration was done using in-line 47 mm filter holders (Sartorius, Inc.) holding 3 sizes of
nylon mesh (1000 µm, 200 µm, 50 µm) followed by a glass fibre filter (Whatman GF/F, 0.8 µm
nominal pore size, muffled before use). These size fractions were intended to collect foraminifera (50-
200 µm) and coccolithophores (0.8-55 µm), and pteropods (200-1000 µm), but the largest size
fraction had insufficient material for analysis. The flow rate at the start of filtration was 25-30 L hour$^{-1}$





and typically dropped during filtration. The 0.8 μm filter was replaced if flow rates dropped below 10
L hour$^{-1}$. Sampling typically took 3 hours. Quantities of filtered seawater were measured using a flow
meter (Magnaught M1RSP-2RL) with a precision of +/-1%. After filtration, remaining seawater in the
system was removed using a vacuum pump. Filters were transferred to 75 mm Petri dishes inside a
flow bench, placed in an oven (SEM Pty Ltd, vented convection) for 3-6 hours to dry at 60 ºC and
stored in dark, cool boxes for return to the laboratory.

A continuous flow Foerst type centrifuge [*Kimball Jr and Ferguson Wood*, 1964], operating at 18700
rpm,  was used to concentrate phytoplankton from the underway  system at a flow rate of 60 L per
hour, measured using a water meter with a precision of +/-1% (Arad). Sampling typically took 1-3
hours. After centrifugation, 500 mL of de-ionized water was run through the centrifuge to flush away
remaining seawater and associated dissolved inorganic carbon.  This was followed by 50 mL of
ethanol to flush away the de-ionized water, ensure organic matter detached from the cup wall, and
speed subsequent drying. Inside a laminar flow clean bench, the slurry in the centrifuge head was
transferred into a 10 mL polypropylene centrifuge tube (Labserve) and the material on the wall of the
cup was transferred using 3 mL of ethanol and a rubber policeman. The sample was then centrifuged
for 15 minutes and 3200 rpm, and the supernatant  (~7 mL) removed and discarded. The vial was
placed in the oven to dry for 12 hours at 60 ºC and returned to the laboratory.

**2.3.2 Distinct analytical procedures for VL1 samples**
POC/PON analyses for the 0.8 μm size fraction collected by filtration were done by packing five 5
mm diameter aliquots (punches) of the 47 mm diameter GF/F filters into acid-resistant 5x8 mm silver
cups (Sercon SC0037), treating these with two 20 μl aliquots of 2 N HCl to remove carbonates [*P*
*King et al.*, 1998], and drying at 60 ºC for at least 48 hours.  For the 50 μm mesh filtration samples,
and the centrifuge samples, 0.5-1.0 mg aliquots of the dried (72 hours at 60 ºC) centrifuge pellet
remaining after PIC coulometry were encapsulated in 4x6 mm silver cups (Sercon SC0036).
Analyses of all these sample types was by catalytic combustion using a Thermo-Finnigan Flash 1112
elemental analyzer calibrated against sulphanilamide standards (Central Sciences Laboratory,
University of Tasmania).  Precision of the analysis was +/- 1 %. A blank correction for of $0.19 \pm 0.09$
ug C was applied which represented 1.6 % of an average sample.

PIC concentrations were determined for subsamples of the 0.8 μm GF/F filters (half of the filter), the
whole 50 μm mesh screens, and the whole centrifuge samples by closed system acidification with HCl
and coulometry using a CM5011 $CO_2$ coulometer.  The samples were placed in glass vials (or in the
case of the centrifuge tubes connected via an adaptor), connected to an acidification unit maintained at
60ºC, acidified with an excess of 2 N HCl, and swept with a nitrogen gas-flow (~100 mL min$^{-1}$) via a
drier into the coulometry cell  Calibration versus calcium carbonate samples provided precision of ±



0.3%. However, for the 0.8 µm filter, precision was limited to 10 % by sub-sampling of the filter due
to uneven distribution. Blank corrections were applied to the 0.8 µm size fraction, being 2.4 ± 1.8 ug
C representing 8.8 % of an average sample.  The 55 µm fraction blank correction was 3.3 ± 0.1 ug C,
representing 22 % of an average sample. Centrifuge pellet coulometry blank subtraction was 2.0 ± 0.1
ug C which was 2.8 % of an average sample.

Biogenic silica analysis of the residues remaining after PIC analysis of the centrifugation samples,
was by vortex mixing, an alkaline digest (0.2 N NaOH) in a 95ºC water bath for 45 minutes, similar to
the method described by Paasche (1973). The samples were then cooled in an ice bath, 1 mL of 1 N
HCl added and mixed, and spun in a bench centrifuge for approximately 10 minutes to remove
undigested solids. 4 mL of each sample was transfered from the centrifuge tubes and filtered using a
syringe filter into a nutrient tube. Six mL of artificial seawater was added to make the sample up to 10
mL. Samples were then analysed using an Alpkem segmented flow analyzer [*Eriksen*, 1997].

**2.3.3 Comparison of VL1 to other voyages**
The first survey on VL1 in 2008 differed from later efforts in two important ways: i) POC and PIC
samples were collected by both filtration and centrifugation, ii) separate BSi samples were not
collected - instead BSi analyses were carried out only on the sample residues from PIC coulometric
sample digestions of the centrifuge samples.  Comparison of POC and PIC results from the
centrifugation samples (effectively total samples without size fractionation) and the filtration samples
(separated into the PIC01 0.8-50 µm and PIC50 50-1000 µm size fractions) shows (Figure 2) that
filtration collected somewhat more PIC (order 20-30 %) and considerably more POC (order 200-300
%) than centrifugation.  This fits with the possibility of loss of material from the continuous
centrifuge cup, with greater loss of lower density organic matter (and possible additional loss of
organic matter via dissolution in the ethanol rinsing step).  Thus for comparison of VL1 POC and PIC
to the other voyages we use only the filtration results, thereby avoiding methodological biases.  For
BSi, we do not have this possibility.  Based on the low centrifuge yields for PIC and POC we can
expect that the VL1 BSi values are also too low.  This is confirmed by comparison to the other
voyages which reveals that VL1 BSi values were lower than those of other voyages, especially in the
far south where BSi values were generally highest (data shown below), but nonetheless had similar
north-south latitudinal trends.  For this reason, our further interpretation of the VL1 BSi results is only
in terms of these latitudinal trends.

**2.4 Analysis of nutrients, DIC, alkalinity, and calculation of pH and calcite saturation**
Nutrients were analysed onboard ship for VL1 to VL5, and on frozen samples returned to land for
VL6-9, all by the CSIRO hydrochemistry group following WOCE/CLIVAR standard procedures,



with minor variations [*Eriksen*, 1997], to achieve precisions of ~1% for nitrate, phosphate, and silicate
concentrations.  Dissolved inorganic carbon (DIC) and alkalinity samples were collected in gas tight
bottles poisoned with mercuric chloride and measured at CSIRO by coulometry and open cell
titration, respectively [*Dickson et al.*, 2007].  Comparison to certified reference materials suggests
accuracy and precision for both DIC and alkalinity of better than ± 2 μmol kg$^{-1}$. Full details were
recently published [*Roden et al.*, 2016].  Calculation of pH (free scale) and calcite saturation were
based on the Seacarb version 3.1.2 software (https://CRAN.R-project.org/package=seacarb), which
uses the default selection of equilibrium constants given in [*Van Heuven et al.*, 2011].

**2.5 Satellite derived ocean properties and the NASA Ocean Biogeochemistry Model**
The locations of oceanographic fronts in the Australian sector were estimated from satellite altimetry,
following the approach of [*S. Sokolov and Rintoul*, 2002], updated as follows.  Absolute sea surface
height (SSH) was calculated by adding the sea surface height anomaly from AVISO+ [*Pujol et al.*,
2016] to the 2500 dbar reference level mean dynamic topography of [*Olbers et al.*, 1992]. The
positions of the fronts were then identified using the sea surface height contours corresponding to the
positions of the Southern Ocean fronts identified by [*S. Sokolov and Rintoul*, 2007a] in the region
100-180 °E. From this analysis, we show 8 fronts from north to south consisting of:
Fronts 1-3) north, middle, and south branches of the SAF, which bound the highest velocity jets of the
ACC.
Fronts 4-6) north, middle, and south branches of the Polar Front, associated with subsurface
temperature features related to the strength of the ACC and with the shoaling of CDW in the
overturning circulation.
Fronts 7-8) north and south branches of the Southern ACC front, marking weaker flows in Antarctic
waters of the ACC and occurring near where upwelling of old nutrient rich and relatively acidic
Circumpolar Deep Water comes closest to the surface.

We do not show the Subtropical Front that marks the northern boundary of the Southern Ocean, or the
Southern Boundary Front, which marks the southern edge of the ACC (separating it from westerly
flow in Antarctic shelf waters).  This is because both features have weak, discontinuous SSH
signatures south of Australia: mesoscale eddies rather than the STF dominate the weak SSH field in
the SAZ, and detection of the Southern Boundary Front is confounded by proximity to the Antarctic
shelf where altimetry is impacted by other processes, including sea-ice cover for much of the year [*S.*
*Sokolov and Rintoul*, 2007a].

We considered using these dynamic heights and front locations as ordinates for the spatial
distributions of POC, PIC and BSi.  In the core of the ACC (50-60 °S), this did help explain some
departures from monotonic north-south trends as resulting as resulting from meanders of the fronts,



but latitude was more strongly correlated with PIC abundance in the SAZ and with BSi in southern
ACC waters and Antarctic shelf waters, where dynamic height contours were only weakly varying.
Accordingly, there was no overall advantage of replacing latitude by dynamic height as a predictor of
biogenic mineral concentrations, and we have used latitude as the ordinate in our figures and
discussion.

Sea surface temperatures (°C) were obtained from the NASA MODIS Aqua 11 μm night-only L3m
product available on-line:
https://giovanni.gsfc.nasa.gov/giovanni/#service=TmAvMp&starttime=&endtime=&data=MODISA_
L3m_SST_2014_nsst&variableFacets=dataFieldMeasurement%3ASea%20Surface%20Temperature
%3B
We chose the night values to avoid shallow ephemeral structures arising from daytime solar heating.
We refer to these estimates simply as SST values.

Phytoplankton chlorophyll concentrations (Chl in mg m$^{-3}$ = ug L$^{-1}$) were obtained from the NASA
MODIS Aqua L3m product available on-line:
https://giovanni.gsfc.nasa.gov/giovanni/#service=TmAvMp&starttime=&endtime=&data=MODISA_
L3m_CHL_2014_chlor_a&variableFacets=dataFieldMeasurement%3AChlorophyll%3B
The full citation for this data is:
NASA Goddard Space Flight Center, Ocean Ecology Laboratory, Ocean Biology Processing Group.
Moderate-resolution Imaging Spectroradiometer (MODIS) Aqua Chlorophyll Data; 2014
Reprocessing. NASA OB.DAAC, Greenbelt, MD, USA.
doi:10.5067/AQUA/MODIS/L3M/CHL/2014.
The algorithm relies on the blue/green reflectance ratio for Chl values above 0.2 ug L$^{-1}$ and
incorporates stray light correction based on the difference between red and blue light reflectances at
lower Chl levels. This product has been suggested to underestimate chlorophyll in the Southern Ocean
south of Australia (Johnson et al., 2013), but has the advantage of ongoing ready availability.  For this
reason, we use it only for context and not for any detailed comparisons to shipboard observations.  We
refer to these estimates as SChl values.

Particulate inorganic carbonate concentrations (mol m$^{-3}$) based on backscatter magnitudes [*W M Balch*
*et al.*, 2005] were obtained from the NASA MODIS/AQUA ocean colour product available on-line:
https://oceancolor.gsfc.nasa.gov/cgi/l3/A20111212011151.L3m_MO_PIC_pic_9km.nc.png?sub=img
The full citation for this data is:
NASA Goddard Space Flight Center, Ocean Ecology Laboratory, Ocean Biology Processing Group.
Moderate-resolution Imaging Spectroradiometer (MODIS) Aqua Particulate Inorganic Carbon Data;





2014 Reprocessing. NASA OB.DAAC, Greenbelt, MD, USA. doi:
10.5067/AQUA/MODIS/L3M/PIC/2014.
We refer to these estimates as SPIC values. The veracity of these estimates in the Southern Ocean
remains an active area of research.  PIC sampling in the Subantarctic South Atlantic found levels 2-3
times lower than the satellite estimates [*W M Balch et al.*, 2011], and the algorithm also produces
surprisingly high estimates in Antarctic waters, where limited shipboard surveys suggest that
coccolithophore abundances drop strongly (work summarized in Balch et al., 2005).  Our data
provides the most extensive PIC observations for comparison to SPIC values in Antarctic waters yet
available, and is discussed in detail below.

Modeled coccolithophore distributions were obtained from the data-assimilating general circulation
model NASA Ocean Biogeochemical Model (NOBM) available on-line:
https://giovanni.gsfc.nasa.gov/giovanni/#service=TmAvMp&starttime=&endtime=&data=NOBM_M
ON_R2014_coc&variableFacets=dataFieldDiscipline%3AOcean%20Biology%3BdataFieldMeasure
ment%3APhytoplankton%3B
The phytoplankton function type model is based on [*Watson W Gregg and Casey*, 2007a].  Details of
particular relevance to comparisons with our observations are discussed in section 3.4.

**3. Results and Discussion**

**3.1 Representativeness of oceanographic sampling**
As shown in Figure 1, sampling covered all Southern Ocean zones from sub-tropical waters in the
north to seasonally sea-ice covered waters in the south (covering SST ranging from -1 to 23 °C).
Almost all samples were representative of high-nutrient low-chlorophyll Southern Ocean waters,
indicative of iron limitation.  Exceptions occurred near Tasmania, where moderate levels of SChl
were occasionally present, and over the Antarctic shelf where locally very high levels of SChl were
present.  Individual maps for each voyage leg of SChl are provided in the Supplementary Material and
of satellite reflectance based estimates of PIC (SPIC) below, and reveal that higher values of SChl and
SPIC are often associated with mesoscale structures, especially in the Subantarctic and Polar Frontal
Zones.  This means that mesoscale variability makes satellite versus shipboard comparisons difficult,
and this problem is exacerbated by frequent cloud cover.  Both techniques characterize the very upper
water column, with ship samples from ~4m depth and the satellite ocean colour observations
reflecting the e-folding penetration depth of ~10-15 m [*Grenier et al.*, 2015; *Morel and Maritorena*,
2001].

It appears likely that our single-depth sampling can be considered as representative of upper water
column phytoplankton concentrations, because pigment samples and profiles of beam attenuation and



night-time fluorescence from some of these voyages as well as previous work show that biomass is
generally well mixed in the upper water column, and that when subsurface chlorophyll maxima are
present they primarily reflect increased chlorophyll levels rather than increased phytoplankton
abundances [*Andrew R. Bowie et al.*, 2011a; *A.R. Bowie et al.*, 2011b; *Parslow et al.*, 2001; *Rintoul*
*and Trull*, 2001; *Shadwick et al.*, 2015; *Trull et al.*, 2001b; *S. W.  Wright et al.*, 1996; *S.W. Wright and*
*van den Enden*, 2000].  This perspective is also consistent with the limited information on the depth
distributions of coccolithophores in the Southern Ocean, which generally exhibit relatively uniform
and maximal values (especially for the most abundant species, *Emiliania huxleyi*) within the surface
mixed layer [*Findlay and Giraudeau*, 2000; *Holligan et al.*, 2010; *Mohan et al.*, 2008; *Takahashi and*
*Okada*, 2000]. There is some evidence that this conclusion can also be applied to the PIC50
foraminiferal fraction, in that the most abundant of these organisms tend to co-locate with
phytoplankton in the mixed layer in the Southern Ocean [*Mortyn and Charles*, 2003].


**3.2 Latitudinal distributions of BSi, PIC, and POC**
All the Voyage Legs exhibited similar latitudinal variations of the measured chemical components
(Figure 3). BSi, predominantly derived from diatoms, was clearly the dominant biogenic mineral in
the south in Antarctic waters.  PIC01 concentrations, predominantly derived from coccolithophores,
were highest in northern Subantarctic waters, although even there BSi was often present at similar
levels.  Interestingly, PIC50 concentrations, predominantly derived from foraminifera, often exhibited
maxima in the middle of the Southern Ocean at latitudes of 55-60 °S. The latitudinal variations in all
these biogenic mineral concentrations were quite strong, exceeding two orders of magnitude.  In
contrast, variations in POC were 10-fold smaller, and often quite uniform across the central Southern
Ocean, with maxima sometimes in the far north near Tasmania and sometimes in the far south over
the Antarctic shelf (Figure 3).  Variations in BSi, PIC, and POC concentrations among the voyages, at
a given latitude, were smaller than these north-south trends.  It seems likely that these smaller
variations were partly seasonal, in that the earliest seasonal voyage leg (VL4 in September) had lower
concentrations of every component.  But across the other voyages, ranging from mid-November
(VL5) to mid-April (VL1) no clear seasonal cycle was exhibited, perhaps owing to variations in
sampling location, and the known importance of inter-annual and mesoscale structures in Southern
Ocean phytoplankton distributions (e.g. [*Moore et al.*, 1999; *Moore and Abbott*, 2002; *S.  Sokolov and*
*Rintoul*, 2007b]). As noted in the Methods section (2.3), the BSi values for VL1 stand out as being too
low, in that they were well below those of other voyages, while the POC, PIC01, and PIC50 values
were similar.

The latitudinal dependence of the relative importance of diatoms and coccolithophores is revealed by
viewing the BSi/PIC01 ratios as an ensemble for all the voyages (use of the ratio helps to remove





seasonal and interannual variations in their abundances which tend to track each other at a given
latitude). The BSi/PIC01 ratio reaches values of 200 in the far south and decreases north of 50 °S to
values near 1 (Figure 4a).  Approximate equivalence of BSi and PIC01 occurs relatively far north in
the Southern Ocean, near 50 °S, and thus near the southern edge of the Subantarctic Zone.  This
persistence of the importance of diatoms as a major component of the phytoplankton community in
northern waters of the Southern Ocean must reflect the winter-time renewal of silica supply from
upwelled deep waters in the Southern Ocean that are carried north by Ekman transport, combined
with recycling of biogenic silica within surface waters, given that by mid-summer silicate is largely
depleted north of the Subantarctic Front [*Nelson et al.*, 2001; *Trull et al.*, 2001b].   Accordingly the
relative dominance of diatoms and coccolithophores in the SAZ may be quite sensitive to changes in
the overturning circulation and westerly wind field.  How this might translate into impacts on the
biological carbon pump remains far from clear.  Interestingly, deep ocean sediment traps in the SAZ
south of Australia reveal strong dominance (4-fold) of PIC over BSi in the export flux to the ocean
interior, reminding us that export can be selective (and also that foraminifera can contribute a
significant fraction of total PIC, estimated to vary from ~1/3 to 2/3; [*A L King and Howard*, 2003]).
The POC flux recovered by these deep sediment traps was close to the global median and similar to
that of biogenic silica dominated fluxes in the Polar Frontal Zone to the south [*Trull et al.*, 2001a].

The importance of diatoms across the entire Southern Ocean, relative to coccolithophores is further
emphasized by expressing their biogenic mineral abundances in terms of associated POC, using
average values for the POC/BSi ratio of iron-limited diatoms (3.35, equivalent to a Si/N ratio of 2 and
Redfield C/N ratio of 6.7 [*Ragueneau et al.*, 2006; *Takeda*, 1998]) and the POC/PIC ratio of
coccolithophores (0.833, for *Emiliania huxleyi*, the dominant Southern Ocean species, [*Bach et al.*,
2015; *M. N. Muller et al.*, 2015]).   As shown in Figure 4b, this suggests that diatoms dominate the
accumulation of organic carbon throughout the Southern Ocean, with coccolithophores generally
contributing less than half that of diatoms in the SAZ and less than a tenth of that in Antarctic waters.
Figure 4b also emphasizes that total POC contents can be largely explained by diatom abundances in
Antarctic waters (south of 50 °S), whereas in the SAZ (north of 50 °S), total POC often exceeds the
sum of contributions from diatoms and coccolithophores.  This serves as an important reminder that
other organisms are important to the carbon cycle in the SAZ, and phytoplankton functional type
models should avoid over-emphasis on diatoms and coccolithophores just because they have
discernable biogeochemical impacts (on silica and alkalinity, respectively) and satellite remote
sensing signatures [*Hood et al.*, 2006; *Moore et al.*, 2002].  Finally, we note that the relatively low
abundance of pelagic calcifying organisms across the Southern Ocean as observed here means that
POC/PIC ratios are high, greater than 4 in the SAZ and ranging up to 20 in Antarctic waters (Figure
4a).  This suggests calcification has a negligible countering impact on the reduction of $CO_2$ partial
pressure by phytoplankton uptake, and thus in mediating $CO_2$ transfer from the atmosphere into the



surface ocean, even smaller than the few to ~10% influence identified earlier from deep sediment trap
compositions in HNLC [*P. W. Boyd and Trull*, 2007a] and iron-enriched waters, respectively [*Salter*
*et al.*, 2014].

Notably, our Southern Ocean PIC01 estimates are smaller than those found in northern hemisphere
polar waters.  As compiled by Balch et al. (2005), concentrations were 100-fold higher (~10 uM) in
the north Atlantic south of Iceland (60-63 °N) than any of our values, and 1000-fold higher than our
values in the same southern hemisphere latitude range.  Values collected over many years from the
Gulf of Maine [*W M Balch et al.*, 2008] were ~ 1 uM, and thus 5-10 times  higher than our SAZ
values (Gulf of Maine summer temperatures are similar to the SAZ, and colder in winter).   This
difference between hemispheres is also evident in observations from the South Atlantic, where PIC
values estimated from acid labile backscatter for 6 voyages between 2004 and 2008 and latitudes 40-
50 °S were  ~0.1-0.5 μM in remote waters [*W M Balch and Utgoff*, 2009], increasing to 1-2μMin the
Argentine Basin with a few values reaching 4μM[*W Balch et al.*, 2014].  These high South Atlantic
observations are the highest of the "Great Calcite Belt" identified as a circumpolar feature of
Subantarctic waters based on SPIC values [*W Balch et al.*, 2014; *W M Balch et al.*, 2011].  Notably,
shipboard PIC measurements in this feature are 2-3 times lower than the SPIC estimates in the South
Atlantic [*W M Balch et al.*, 2011], and ship collected samples from two voyages across the South
Atlantic and Indian sectors [*W M Balch et al.*, 2016]  exhibit PIC concentrations (actual PIC values
accessed online at http://www.bco-dmo.org/dataset/560357, rather than the PIC estimates from acid-
labile backscatter shown in the paper) that decrease eastwards in this feature to reach values close to
our observations in the Australian sector of ~ 0.1 μM(Figure 3).

**3.3 Comparison to satellite PIC (SPIC) estimates**
As is very evident from the limited observations we have achieved from our efforts over many years,
it will never be possible to characterize Southern Ocean phytoplankton population dynamics from
ship based sampling – the influences of mesoscale circulation, ephemeral inputs of the limiting
nutrient iron, and food web dynamics produce variability that cannot be adequately assessed in this
way, leaving sparse sampling open to potentially large biases.  Use of satellite observations is clearly
the path forward to alleviate this problem, and development of algorithms for global coccolithophore
distributions has been a major advance [*W M Balch et al.*, 2005; *Brown and Yoder*, 1994].  Until
recently the calibration of these SPIC values has been based primarily on North Atlantic observations.
Work to check these efforts for the Southern Ocean has begun, but remains sparse.  Early work in the
South Atlantic found that SPIC values appeared to exceed in ocean PIC by a factor of 2-3 [*W M Balch*
*et al.*, 2011], and based on a handful of samples it was suggested that this might reflect a lower
amount of PIC per coccolith [*Holligan et al.*, 2010].  Two dedicated voyages to investigate the "Great



Calcite Belt" in the SAZ and PFZ across the South Atlantic and South Indian Oceans, attempted
comparison of acid-labile backscatter (as a proxy for PIC) and MODIS SPIC values, but there were no
match-ups in the South Atlantic owing to cloudy conditions [*W M Balch et al.*, 2016]. Results from
the South Indian sector, and from other voyages in the South Atlantic show high acid-labile
backscatter which translates into high SPIC estimates in the SAZ and PFZ  (especially in naturally
iron-fertilized waters), but also high values further south which are not in agreement with ship
observations [*W M Balch et al.*, 2016; *Smith et al.*, 2017].

Comparsion of our ship observations to MODIS SPIC estimates are shown in Figure 5 for each
voyage leg.  These reveal some agreement in the SAZ in terms of identifying moderate levels of PIC,
often in association with higher levels of total SCHL (Supplementary Material), but differ strongly in
Antarctic waters where all ship observations reveal low PIC values, whereas the SPIC estimates in
Antarctic waters reach and often exceed those in the SAZ, especially over the Antarctic shelf.  Our
sparse data do not permit a comparison in the SAZ sufficient to quantify possible differences between
the SPIC and PIC values there (only ~20 cloud-free match-ups were achieved, and about half of these
in waters with very low PIC), but are in rough agreement with the earlier estimate of an over-
estimation by the satellite algorithm of a factor of 2-3 [*W M Balch et al.*, 2011].

**3.4 Comparison to possible environmental controls on coccolithophore growth rates**
The ship observations provided here offer a significant advance in quantifying the distributions of
coccolithophores in the Southern Ocean south of Australia, but much less understanding of why these
distributions arise and therefore how they might change in response to climate, circulation, and
biogeochemical changes in the future.  Coccolithophores, especially the most common species
*Emiliania huxleyi*, have been studied sufficiently in the laboratory to allow possible important
controls on their niches and especially their calcification rates to be proposed, including temperature,
pH, pCO$_2$, calcite saturation state, and macro- and micro-nutrient availability [*Bach et al.*, 2015; *Feng*
*et al.*, 2016; *Mackinder et al.*, 2010; *M. N. Muller et al.*, 2015; *M.N. Muller et al.*, 2017; *Schlüter et*
*al.*, 2014; *Schulz et al.*, 2007; *Sett et al.*, 2014]. We collected observations of many of these properties
in parallel with our PIC observations, and now briefly examine whether they present correlations that
might contribute to understanding why coccolithophores are found mainly in northern Subantarctic
waters, and not further south.  For illustrative purposes, we focus on VL3 (the mid- to late summer I9
northward hydrographic section from Antarctica to Perth) and VL6 (the early to mid-summer
southward Astrolabe transit from Tasmania to Antarctica). VL3 covered the widest range of physical
properties, and exhibited PIC01 concentrations that remained elevated further south than any other
voyage (Figure 3).  VL6 exhibited the more typical PIC01 distribution of a close to continuous
decrease southward (Figure 3). The results from the other Voyage Legs were very similar to VL3
(figures not shown; data available in Supplementary Materials)..




Many properties that might influence coccolithophore productivity decreased strongly and close to
monotonically from north to south across the Southern Ocean for our voyages (Figure 6). These
include temperature (from 23 to -0.4 C for our samples), salinity (from 35.6 to 33.6, with tight
correlation with alkalinity, not shown - data available in the Supplementary Material), pH (from 8.20
to 8.08 on the free scale), and the saturation state of calcite (from 5.22 to 2.12).  The strong
correlation of these properties means that it is not easy to separate their possible influences on
coccolithophore distributions, without relying on specific thresholds or quantitative response models.
With the added complexity of a lack of information on individual species, or the availability of iron as
the limiting micro-nutrient, deducing a possible influence of ocean acidification on coccolithophore
distributions from our spatial distribution data is very difficult, and well beyond our scope.
Nonetheless, we offer a few pertinent observations.  Firstly, the change in PIC01 abundances with
latitude is much larger than expected from models of the responses of calcification rates (normalized
to maximum rates) to inorganic carbon system variations (Figure 6).  Two models are shown:

The "Bach model" based on independent terms for sensitivity to bicarbonate, $CO_2$, and pH. It fits
quite well the results from many laboratory incubations of *Emiliania Huxleyi* strains under conditions
of modern and elevated pCO$_2$ [*Bach et al.*, 2015], and we have used values for the constants (a, b, c,
d) obtained from incubations of a strain isolated from Subantarctic waters south of Tasmania [*Müller
et al.*, 2017] to provide what might be considered the best current model for the calcification rate
response to changing inorganic carbon abundance and speciation, following Eq. (1):

Bach relative calcification rate = $a\ [HCO_3^-]\ /\ (b+[HCO_3^-]) - e^{-c[CO2]} - d[H^+]$         (1)


The "Langdon model" based on a simple, inorganic precipitation motivated parameterization of
calcification as a function of calcite saturation state $\Omega$ [*Gattuso et al.*, 1998; *Langdon et al.*, 2000],
which has been shown to apply in an approximate way to many corals [*Anthony et al.*, 2011;
*Silverman et al.*, 2007], and perhaps to Southern Ocean foraminifera [*Moy et al.*, 2009]. We have
chosen the simple linear form (n=1) and a sensitivity at the top end of the observed range (a =1/4, so
that calcification rate varies linearly from 0 to 1 for $\Omega$=1 to 4), following Eq. (2):

Langdon relative calcification rate = $a\ (\Omega-1)^n$         (2)



As shown in Figure 6, both these calcification rate models exhibit limited variations with latitude in
the Southern Ocean.  The Bach model suggests negligible change in calcification rate.  This is



essentially because the Southern Ocean variations in bicarbonate, $CO_2$, and pH are very small
compared to the future expected values used in incubation experiments. In addition, southward
cooling causes pH to rise, offsetting the impact of southward decrease in salinity and alkalinity, thus
reducing the southward decrease of pH and the associated drop in modeled calcification rate. The
Langdon model suggests approximately 3-fold decrease in calcification rate, which is considerably
smaller than the more than 10-fold drop in PIC01 (shown on a linear scale in Figure 6 and a
logarithmic scale in Figure 3). The shape of the Langdon model decrease shows some agreement with
that of PIC01 for VL6, but none for VL3 (which exhibits relatively constant significant PIC01
concentrations in the 40-50 °S latitude range where the Langdon model shows a strong decrease in
calcification rate, and then a strong drop in PIC01 south of 60 °S where the Langdon model shows no
change). Thus, and unsurprisingly, coccolithophore abundances are clearly not controlled by
inorganic carbon chemistry alone.

Many laboratory studies have emphasized the importance of temperature on coccolithophore growth
rates, as compiled recently [*Feng et al.*, 2016], and warming has been suggested as a possible cause of
decadal northward apparent range expansion in the North Atlantic [*Rivero-Calle et al.*, 2015] and the
occurrence of unusual blooms in the Bering Sea [*Merico et al.*, 2004]. To provide a brief
visualization of the expected univariate response, we fit the "Norberg" thermal optimum envelope
model [*Norberg*, 2004] to growth rate data for 5-25 °C with modern $pCO_2$ and nutrient replete
conditions for a Southern Ocean morphotype A strain of *Emiliania Huxleyi*, isolated from south of
Tasmania [*M. N. Muller et al.*, 2015], with optimum temperature z=15, thermal window w=10, and
scaling constant *a*, in which the exponential term represents the broad global temperature dependence
of generic phytoplankton growth rates [*Eppley*, 1972] and produces the known skewed form of
organismic thermal tolerances, following Eq. (3):

$$\text{Norberg growth rate (d}^{-1}) = a\,[1-((T-z)/w)^2]\,e^{0.0633T} \qquad (3)$$

As shown in Figure 6, this predicts a drop from ~0.5 d$^{-1}$ at the northern edge of the Southern Ocean to
zero growth near ~53 °S, whereas PIC01 concentrations fall off more slowly further south. The
presence of other morphotypes with lower thermal optima [*Cubillos et al.*, 2007] is an easy possible
way to explain this difference. Overall the Norberg temperature model has an advantage of the
calcification rate models – it does predict a strong decrease to negligible PIC01 values in the south.
There are of course many other possible explanations.

Interestingly, these uncertainties regarding the roles of inorganic carbon chemistry and temperature on
Southern Ocean coccolithophore distributions contrast with the possible role of macro-nutrients, in
that phosphate and nitrate increase southward across the Southern Ocean (e.g. [*Trull et al.*, 2001b]),





and were everywhere abundant during our surveys (nitrate > 3 uM, with phosphate/nitrate close to
Redfield expectations, data in Supplementary Material), and thus would be expected to lead to
southward increases in coccolithophore abundances which were not observed. For this reason we
suggest nitrate and phosphate availability is not an obvious driver of the southward decrease in
coccolithophore abundances in Southern Ocean HNLC waters (i.e. these nutrients are sufficient
everywhere), although these nutrients may be important in determining the success of
coccolithophores in oligotrophic waters at the northern edge of the Southern ocean, given the high
half-saturation constant for nitrate uptake observed in some laboratory studies (~13 uM; [*Feng et al.*,
2016]), and the possibility that high temperature and low nutrient conditions may non-linearly amplify
phytoplankton stresses [*Thomas et al.*, 2017].

Importantly, in addition to multivariate environmental control of coccolithophore distributions via
their growth rates, there is the possibility of control by resource competition with other autotrophs
(presumably mainly for iron) and/or stronger loss terms to grazers in Antarctic than Subantarctic
waters. These are difficult issues to evaluate, and we provide just one comment. Diatom abundances
as estimated from BSi concentrations show a stronger latitudinal relationship to silicon availability
than coccolithophores do to carbonate availability (Figure 6). Diatoms abundances drop strongly near
the SAF, north of which summer time Si(OH)$_4$ concentrations drop below 1 uM, i.e. close to the
'residual" concentration which it appears diatoms cannot access [*Paasche*, 1973]. Surveys of
coccolithophores and diatoms in the SAZ in the South Atlantic and South Indian sectors have
previously suggested that coccolithophore distributions may be linked to competition with diatoms [*W
M Balch et al.*, 2016; *Smith et al.*, 2017], and this view is compatible with our observations, although
it remains unproven. Further progress in understanding the controls on coccolithophore abundances in
the Southern Ocean is clearly needed. At present temperature and competition with diatoms for iron
appear to be the strongest candidates (at least for southward expansion; with nitrate a strong influence
on the location of the northern oligotrophic boundary; [*Feng et al.*, 2016]).

**3.5 Comparison to the NASA Ocean Biogeochemical Model**
Many of these ideas about the roles of environmental conditions and ecological competition have
been included in models for global coccolithophore distributions, e.g. [*Watson W Gregg and Casey*,
2007a; *Le Quere et al.*, 2005]; and we provide a brief comparison to one model – the NASA Ocean
Biogeochemical Model (NOBM) for which simulation results are available on-line (see the Methods
section). In brief, the NOBM predicts coccolithophore abundances (in Chl units) that are restricted to
the far northern reaches of the Southern Ocean (Figure 7). This is also true for the Dynamic Green
Ocean Model [*Le Quere et al.*, 2005]. This contrasts with our PIC results (Figures 3, 4, 6) and with
PIC and coccolithophore cell counts from other sampling efforts which have found coccolithophore
abundances to extend with similar concentrations right across the SAZ and sometimes the PFZ, e.g.



during VL6 south of western Australia (Figures 3 and 6), south of Tasmania [*Cubillos et al.*, 2007],
in the Scotia Sea [*Holligan et al.*, 2010], and in the South Atlantic and South Indian Oceans,
especially in regions of natural iron fertilization [*W M Balch et al.*, 2016; *Smith et al.*, 2017]. In the
NOBM, diatoms are also simulated and show (Figure 7) the expected high abundance in Antarctic
waters in the southern third of the Southern Ocean, decreasing northward as in our results (but also
show a band of elevated diatom concentrations in the Subantarctic, which we did not observe).

Competition for nutrients in the NOBM favours the ability of coccolithophores over diatoms to get by
on limited resources (half-saturation constants for nitrate and iron of 0.5 and 0.67 versus 1.0 and 1.0
uM) including light (half saturation constant of 56 versus 90 umol photons m$^{-2}$ s$^{-1}$ under Southern
Ocean low light conditions). But diatoms are specified to have higher growth rates when all resources
are non-limiting (maximum growth rate at 20 °C 1.50 versus 1.13, both with the same Eppley
dependence on temperature). Thus in the model, diatoms dominate silicon replete Southern Ocean
waters, outcompeting other species for the limiting iron, and only give way to other species when
silicon is depleted. Notably these other species then do best when additional Fe is supplied from
either atmospheric sources (in the north where continental dusts are not shielded by ice) or island
oases such as Crozet or Kerguelen. This view is compatible with our observations and those carried
out in the northern half of the Southern Ocean during the "Great Calcite Belt" voyages [*W M Balch et*
*al.*, 2016; *Smith et al.*, 2017]. It suggests that potential expansion of coccolithophores southward
might be linked to decreasing supply of silicon from reduced upwelling of Circumpolar Deep Water
in a progressively more stratified global ocean. A cautionary note to this conclusion is provided by
the NOBM simulation of significant concentrations of diatoms in the SAZ where silicon is low, which
arises from their specified higher maximum growth rate, emphasizing the importance of this
parameter, and its temperature dependence, in modeling phytoplankton distributions. In specifying
this temperature dependence, this model and most others still rely on the global compilation from
nearly 50 years ago [*Eppley*, 1972]. Clearly better understanding of the controls on maximum growth
rates and their temperature tolerance for key phytoplankton taxa is needed, first to understand current
distributions and then to explore possible future changes.






**4. Conclusions**
Our surveys of PIC concentrations as a proxy for coccolithophores in the Southern Ocean south of
Australia suggest:




- The concentrations of coccolithophores were much smaller (at least 10-fold) in the open
  Southern Ocean south of Australia than in northern hemisphere oceans.

- Coccolithophores were most abundant in the Subantarctic Zone, and occasionally in the Polar
  Frontal Zone.

- The contribution of coccolithophores to total phytoplankton biomass (estimated from POC)
  was small, less than 10% in Subantarctic waters and less than 1% in Antarctic waters.

- The "Great Calcite Belt" characterization of SAZ and PFZ waters based on satellite estimates
  of PIC (SPIC) is overstated south of Australia.  The SPIC estimates appear to be too high by a
  factor of 2-3 in the SAZ, and given their low contribution to total PIC it does not appear that
  coccolithophores have a dominant role regional marine ecology.

- Even greater care must be taken in the use of satellite PIC (SPIC) estimates south of the
  Subantarctic Front, because the algorithms erroneously identify large agglomerations of PIC
  where none is present south of Australia.

- Our PIC results and ancillary measurements of biogenic silica, particulate organic carbon,
  dissolved nutrients, and inorganic carbon system status may be useful in the testing of models
  of limiting conditions and ecological competitions that affect coccolithophore distributions.
  Preliminary considerations suggest that temperature, iron, and competition with diatoms may
  be stronger influences than pH or calcite saturation state.

Despite the considerable effort required to obtain these survey results, much remains to be done just to
define coccolithophore distributions, for example their seasonality, especially when the complexities
of differing responses of individual species and strains are considered.



*Acknowledgements*
We thank Steve Rintoul (CSIRO Oceans and Atmosphere, Hobart) and Alain Poisson (Universite
Pierre et Marie Curie, Paris) for allowing sample collection to proceed under the auspices of their
science programs onboard *Aurora Australis* and *l'Astrolabe*, respectively.  ACE CRC staff, students,
and volunteers carried out onboard sampling, including Peter Jansen, Stephane Thannassekos,
Elizabeth Shadwick, and Nick Roden. Nutrient analyses were carried out by the CSIRO
hydrochemistry group. Thomas Rodemann at the UTAS Central Sciences Laboratory did the CHN
analyses.  Funding was provided by the Australian Commonwealth Cooperative Research Centre
Program via the ACE CRC.  Tim Smit participated in the first voyage in 2008 and described those
results in his Utrecht University Masters thesis and in a poster presented at the Second Symposium on
the Ocean in a High $CO_2$ World, Monoco, October 6-9, 2008. Andrew Lenton (CSIRO) produced the
database of absolute mean dynamic heights and associated front locations.






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




**Table 1. Sample Collection**

| #   | Voyage Name           | Leg   | Dates                | PIC50[3] | PIC01 | POC  | BSi  |
| --- | --------------------- | ----- | -------------------- | -------- | ----- | ---- | ---- |
| VL1 | AA2008_V6 (SR3)       | North | 28/03/08–15/04/08    | 57/0     | 59/0  | 59/0 | 59/0 |
| VL2 | AA2012_V3 (I9)        | South | 05/01/12–20/01/12[1] | 4/16     | 4/16  | 9/25 | 7/22 |
| VL3 | AA2012_V3 (I9)        | North | 20/01/12–09/02/12    | 62/0     | 62/0  | 59/0 | 53/0 |
| VL4 | AA2012_VMS (SIPEXII)  | South | 13/09/12–22/09/12    | 0/21     | 0/20  | 0/24 | 0/24 |
| VL5 | AA2012_VMS (SIPEXII)  | North | 11/11/12–15/11/12    | 0/25     | 0/25  | 0/27 | 0/28 |
| VL6 | AL2013_R2 (Astrolabe) | South | 10/01/13–15/01/13    | 0/25     | 0/25  | 0/23 | 0/25 |
| VL7 | AL2013_R2 (Astrolabe) | North | 25/01/13–30/01/13    | 0/27     | 0/27  | 0/26 | 0/27 |
| VL8 | AA2014_V2 (Totten)    | South | 05/12/14–11/12/14    | 0/36     | 0/36  | 0/32 | 0/37 |
| VL9 | AA2014_V2 (Totten)    | North | 22/12/14–24/01/15[2] | 6/44     | 6/44  | 8/27 | 8/39 |

[1] 18/01/12-20/01/12 east-west traverse from ~ 65º S 144º E to 65º S 113º E included in South leg

[2] 22/12/14-11/1/15 west-east traverse from ~ 65º S 110º E to 65º S 140º E included in North leg

[3] Numbers of samples collected on station / underway



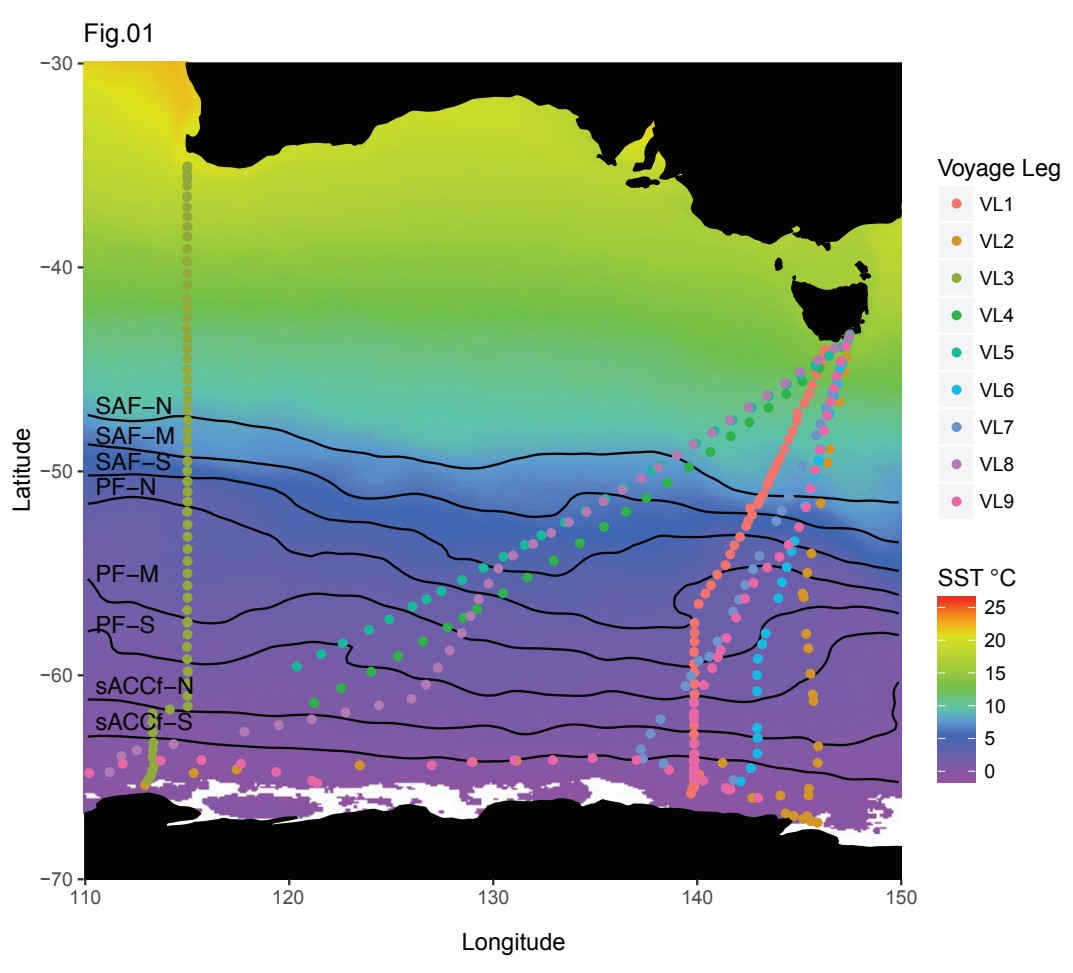





Fig.2

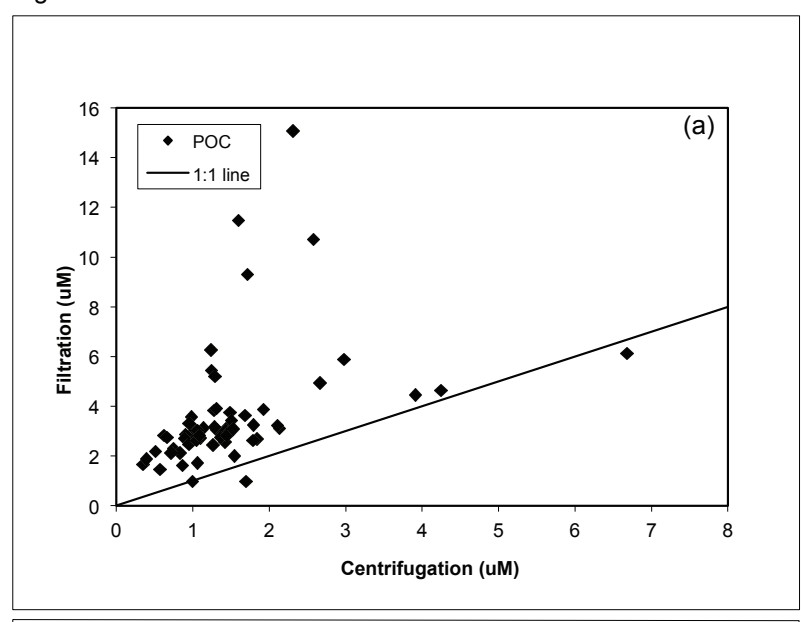

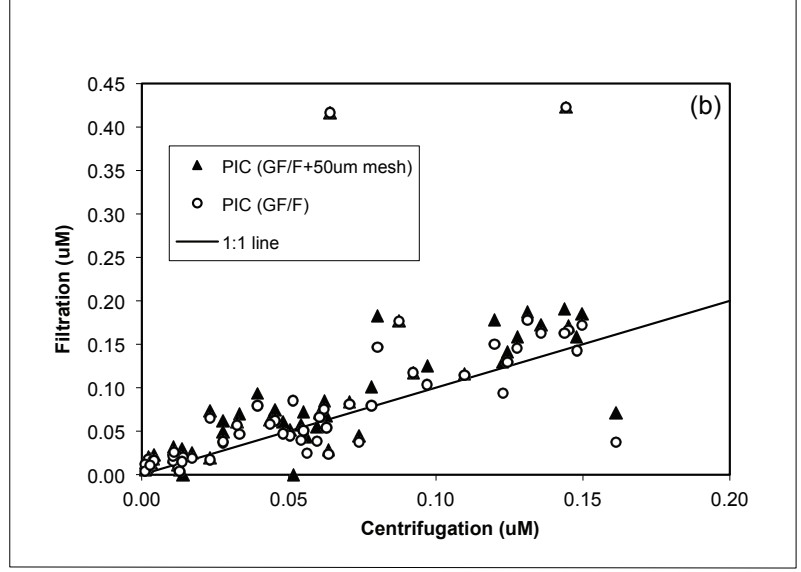





Fig.03




Fig.04

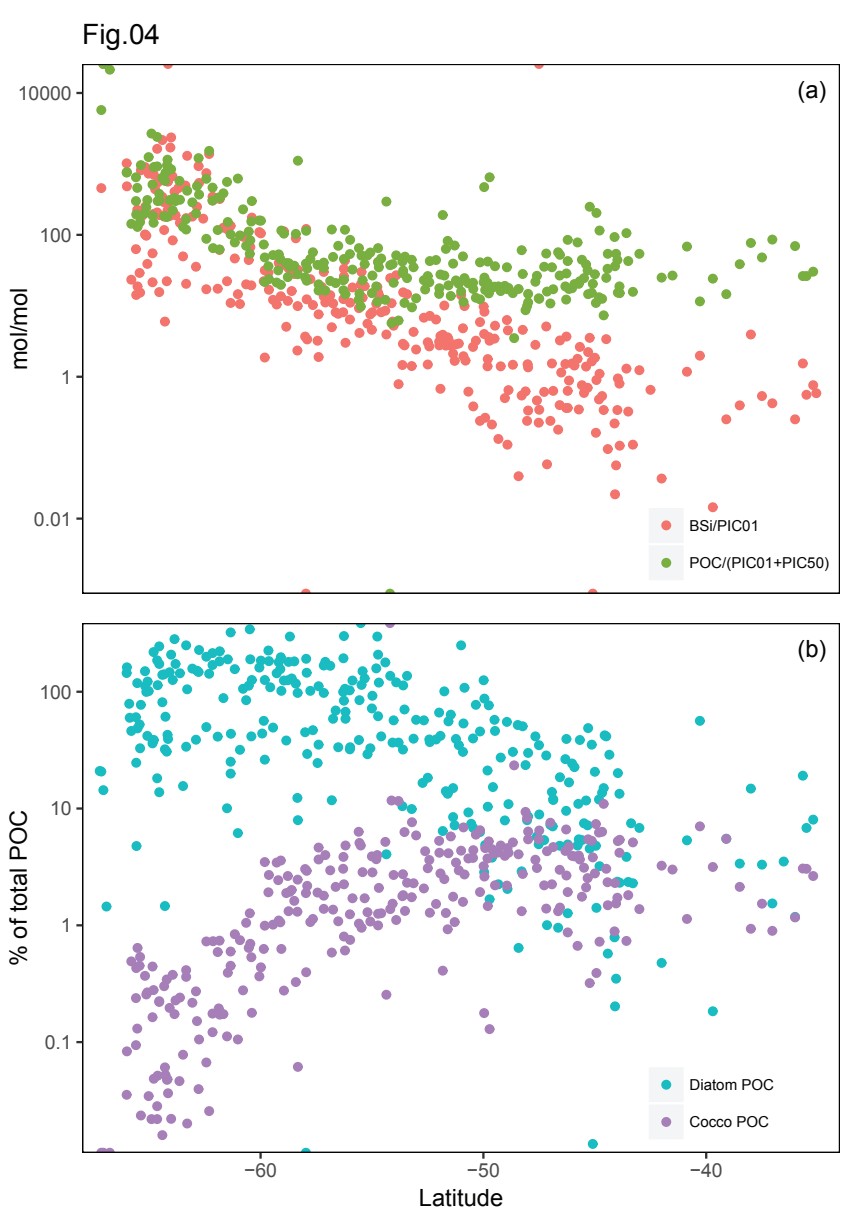



Fig.05



Fig.06



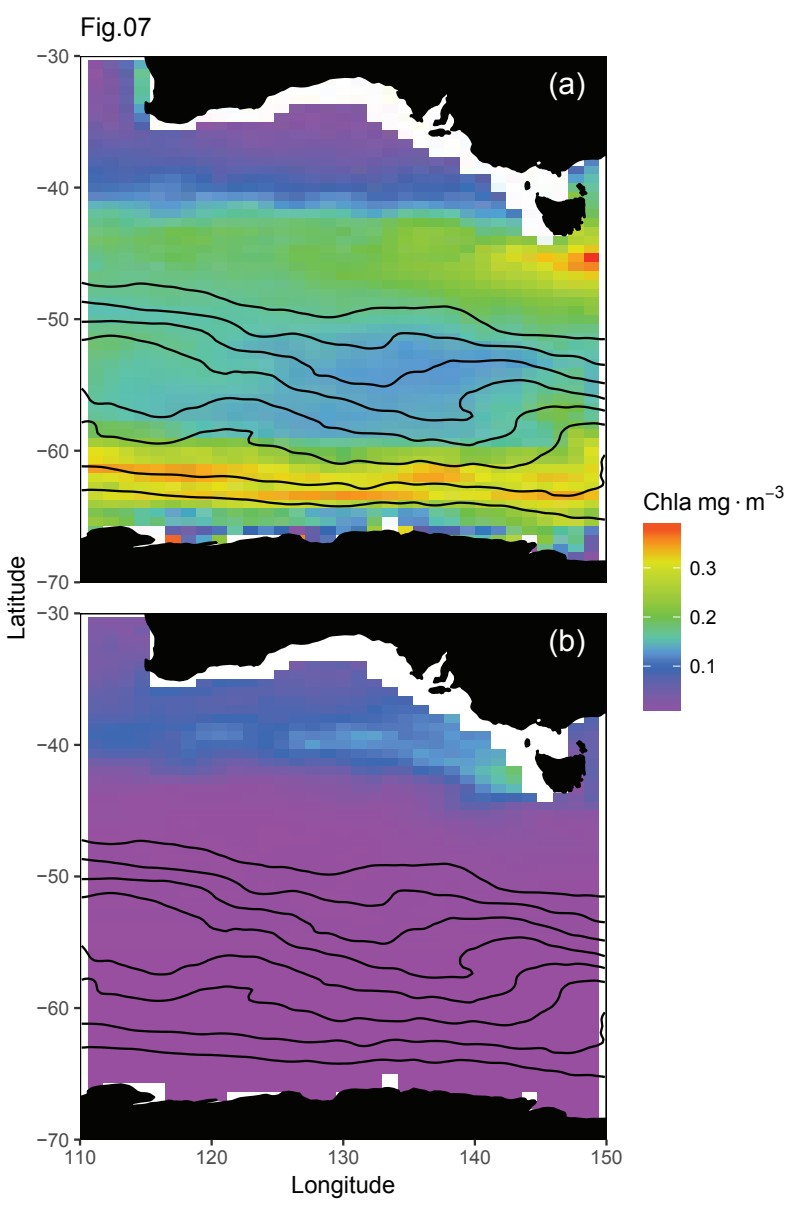