# Peer review of "Australia: a baseline for ocean acidification impact assessment"

_Biogeosciences, 2017_

## Referee Comment (RC1) · L.T. Bach (Referee) · 27 Jun 2017

Review on: "Distribution of planktonic biogenic carbonate organisms in the Southern Ocean south of Australia: a baseline for ocean acidification impact assessment " by Trull et al. In this study, Trull et al., investigate Diatom and calcifier distribution patterns in the Southern Ocean. Their analysis is based on BSi, POC and size fractionated PIC data. They compare their ground truth data with satellite data and model predictions and report important discrepancies and consistencies. I think their study is very valuable and their paper contains key information to document climate change effects on diatoms and calcifiers in the Southern Ocean. I really only have minor comments. Some of these are addressing their methods and some refer to the discussion/conclusion part.

Line 26: Are diatoms really the most abundant phytoplankton? I can understand that they might be dominant in terms of biomass but would intuitively assume that smaller groups (e.g. picoeukaryotes such as Micromonas) are more abundant than diatoms. (I may be wrong here but just to double check.)

Line 56: I am not sure that the under-saturation is primarily due to low TA. I would assume that it is due to the low temperature that leads to generally low carbonate ion concentration.

Line 63: It is a bit weird that you say that their relative importance is poorly known but then in the same sentence say that they will have an influence ecosystem health. The second part of the sentence implicitly contradicts the first part. Furthermore, I did not understand how the "importance" will "influence of the overall impact. . .". This sentence could perhaps be rephrased.

Line 80: Aren't these results? Perhaps move this sentence to results part. Furthermore, I do not understand the use of the second "suggested" in this sentence. Please check.

Line 91: In this context it may also be useful to remind the reader that the PIC50 fraction could also contain aggregated coccolithophore calcite (e.g. within fecal pellets).

Line 152: It would be helpful to know whether or not you expect a loss of CaCO3 by sieving the samples. Are there large quantities of CaCO3 expected in the >1000 $\mu$m size fraction?

Line 152: What do you mean by "ship clean"? Please clarify.

Line 154: Can you provide any information if the 50 $\mu$m filter tended to block when such a large volume is filtered? I am asking because it could be that towards the end

of the filtration process also smaller particles might have been retained on the filter due to clogging. I know this is difficult to reconstruct, but in case you have any further information it would be useful to provide them. I have personally made bad experiences with sequential filtrations.

Line 158: I am a bit nervous about the PIC filter cleaning procedure. Omega is 0 in the deionized water and the pH is (probably) low. Does the deionized water have the potential to dissolve $CaCO_3$?

Line 336: "as resulting" twice.

Line 402: I do not understand why mesoscale variability makes the comparison difficult. If you are at a certain location with a ship and sample PIC and you have satellite data for the very same time, you could easily compare these values, couldn't you?

Line 405: What is "e-folding"? The term has not been introduced.

Line 469: Dominant in terms of abundance? Dominant in terms of biomass would probably be the more important metric here.

Lines 470 and 473 : These results imply that diatoms (and to a limited extent coccolithophores) more or less exclusively contribute to the bulk POC in Antarctic waters. I am not so experienced with the plankton communites in the Southern Ocean but would intuitively disagree. Is it really possible that diatoms are so dominant? What about grazers? Did the analysis include e.g. copepod as a POC source or were these not captured on the filters? I think the result of bulk POC = diatom POC in the Antarctic is very interesting.

Line 480: Abundance of calcifiers or concentration of $CaCO_3$? I think you should stick to the latter term to be more precise.

Line 482: You argue that PIC/POC is low which leads to little influence on the TA-mediated reduction of atmospheric $CO_2$ uptake. I agree with that. However, PIC can induce biogeochemical feedbacks in other ways e.g. through ballasting (as you mention in the paragraph before). So I think that it is not really valid to say that coccolithophores had a limited influence on the uptake capacity of atmospheric CO2 if you neglect other feedback mechanisms than TA reduction.

Section 3.4: In section 3.4 you compare model predictions with field data to test whether they predict meaningful trends. I think this is extremely valuable. I have, however, two comments.

1) You first use the Bach et al., 2015 and Langdon et al. 2000 models. These models only consider carbonate chemistry conditions and no other environmental parameter to predict calcification rates. Your data nicely shows that carbonate chemistry is obviously not the driving factor behind the latitudinal trend in the Southern Ocean because model prediction and latitudinal patterns are inconsistent. The Bach et al., model basically predicts that the carbonate chemistry conditions are close to ideal throughout the Southern Ocean. The Langdon et al., model predicts a decline which reflects the trend in Omega. Both models describe calcification response to carbonate chemistry and not distribution patterns of calcifiers. The reason why I mention this is because at the end of this part of the paper you state: "Thus, and unsurprisingly, coccolithophore abundances are clearly not controlled by inorganic carbon chemistry alone" (Lines 603-604). I could not agree more with this statement. However, the way this is formulated implies to some extent that your finding contradicts what we have concluded in our study. But this is not the case. In Bach et al. (2015) we wrote: "great care must be taken when correlating carbonate chemistry with coccolithophore dispersal because this is by no means the only parameter controlling it. Physical (e.g. temperature), other chemical (e.g. nutrient concentrations), or ecological (e.g. grazing pressure) factors will under many if not most circumstances outweigh the influence of carbonate chemistry conditions, unless differences in the latter are extreme. We will therefore focus the discussion on those cases where differences in carbonate chemistry conditions are rather extreme." Thus, our valuation is very similar to that of the authors of this manuscript. I would appreciate if you could point out that your main conclusion in this

paragraph (that carbonate chemistry is probably not the key factor controlling coccolithophore distribution) is also in line with (and not conflicting with) what we assumed in our studies.

2) I am a bit skeptical about the growth rate vs. temperature argument based on the Norberg model. The model predicts a decline of coccolithophore growth rates due to decreasing temperature. This in itself is not convincing because the decrease of growth rate would apply for every other phytoplankton group as well. What you would really have to look at is if the growth rate of coccolithophores decreases over-proportionally relative to other phytoplankton species. If this was then case, then you could argue that coccolithophores become less competitive the further South you go.

Line 653: In this concluding remark you only consider the bottom-up control on diatom vs. coccolithophore distribution. Have you also considered if top-down mechanisms could have played a role here? Even though there may not be appropriate data available to test this in the present study, it may still be useful to remind the reader that this mechanism exists and could also have played a role. I think the Assmy et al., (2013) study nicely made the case that predators may have an important influence on phytoplankton composition in the Southern Ocean.

Table 1: I think the uppercase 3 also needs to be added to PIC01, POC, and BSi.

Figure 1: It would be helpful to add full names and abbreviations of the various fronts to the figure caption.

Figure 3: One particularly interesting finding presented in Figure 3 is that PIC50 (foraminifera) concentrations are considerably lower than PIC01 (coccolithophores) concentrations except for maybe the most Southern stretch of the transects. Sometimes the discrepancies are orders of magnitude. This implies that coccolithophores are the much more important pelagic calcifiers in the Southern Ocean than foraminifera and pteropods. Is this conclusion valid? If so, I think this finding definitely deserves more attention in your paper. Furthermore, it would be interesting to compare this with

the results from Broecker and Clark (2009) who found roughly equal contribution of coccolithophores and foraminifera to the sediment CaCO3 (although their most southern sample came from 40° South).

Figure 4: What was the rationale of showing the POC/PIC ratio? I think readers will generally be more familiar with PIC/POC ratios.

I hope my comments help to further improve the manuscript.

With kind regards,

Lennart Bach

---

## Referee Comment (RC2) · A.J. Poulton (Referee) · 10 Aug 2017

GENERAL COMMENTS:

This is an interesting, well-written and comprehensive study of the concentration and distribution of bio-mineralised particles over latitudinal sections (40 to 65oS) south of Australia. I commend the authors on the dedication that construction of such a dataset has taken, with cruises from 2008 to 2015 using (mostly) standardised methodology. I have one significant recommendation and a few minor comments.

[Figure]

My recommendation is that the authors need to further elucidate in the manuscript the distinction between particulate material and 'living biomass'. Most particulate pools (POC, PIC and BSi) contain variable contributions of both living biomass (i.e. organisms) and detrital material. In the case of PIC, detached coccoliths (e.g.) can be a significant fraction of the total pool and their small size conveys very slow sinking speeds. In the case of diatoms, empty or broken frustules may stay in suspension (or as part of the cell chain) after the organic material has been removed. Similar comments may be made about the larger biogenic organisms (foram fragments, juvenile shells, radiolarian tests). While the particulate concentrations of PIC and BSi represent well the (historical) production of biogenic material from coccolithophores and diatoms, as well as the other groups examined here, their relationship to 'living biomass' is not necessarily direct and may break down seasonally and spatially. Recognising that this may occur, for example in post-bloom conditions, is an important caveat that should be clear to the reader.

MINOR COMMENTS

Ln 107-108: The calcite content of these different strains of Emiliania huxleyi also differ significantly (see Poulton et al., 2011 for estimates or Muller et al., 2015 for measurements), which may have strong implications for PIC production in S Ocean coccolithophore blooms (e.g. Poulton et al., 2013).

Ln 263: Missing full stop between 'cell' and 'Calibration'.

Ln 468-470: The POC:PIC ratio given is relatively low, especially for the S Ocean strain: Muller et al. (2015) reports values of ~0.83 for over-calcified strains, ~1.5 for normal A-type and greater than ~2 for the B/C type reported in the S. Ocean. Maybe the authors could add in a statement on the sensitivity of their estimates to cell POC:PIC ratios – and also how detached coccoliths may actually counteract high cellular POC:PIC ratios.

Ln 518-519: Biometric measurements have confirmed the low PIC per coccolith for the

different morphotypes/strains (see Poulton et al., 2011 and/or Charalampopoulou et al., 2016; see also Muller et al., 2015 (as cited)).

Ln 573: Please correct Emiliania Huxleyi to Emiliania huxleyi.

Ln 622: Light utilization may be another important factor as there are pigment differences between E. huxleyi strains (see Cook et al., 2011)

Ln 653-655: Charalampopoulou et al. (2016) concluded that temperature and light were strong drivers of coccolithophore distribution and calcification across a latitudinal transect in Drake Passage (whilst also acknowledging the role of iron). Have the authors considered the role of (seasonal) light availability?

References

Cook et al. (2011). Photosynthetic pigment and genetic differences between two Southern Ocean morphotypes of Emiliania huxleyi (Haptophta). Journal of Phycology 47, 615-626.

Poulton et al. (2011). Biometry of detached Emiliania huxleyi coccoliths along the Patagonian Shelf. Marine Ecology Progress Series 443, 1-17, doi: 10.3354/meps09445.

Poulton et al. (2013). The 2008 Emiliania huxleyi bloom along the Patagonian Shelf: Ecology, biogeochemistry, and cellular calcification. Global Biogeochemical Cycles 27, 1-11, doi: 10.1002/2013GB004641.

Charalampopoulou et al. (2016). Environmental drivers of coccolithophore abundance and calcification across Drake Passage (Southern Ocean). Biogeosciences 13, 5917-5935, doi: 10.5194/bg-13-5917-2016.

---

## Author Comment (AC1) · 11 Oct 2017

My recommendation is that the authors need to further elucidate in the manuscript the distinction between particulate material and 'living biomass'. Most particulate pools (POC, PIC and BSi) contain variable contributions of both living biomass (i.e. organisms) and detrital material. In the case of PIC, detached coccoliths (e.g.) can be a significant fraction of the total pool and their small size conveys very slow sinking speeds. In the case of diatoms, empty or broken frustules may stay in suspension (or as part of the cell chain) after the organic material has been removed. Similar comments may

be made about the larger biogenic organisms (foram fragments, juvenile shells, radiolarian tests). While the particulate concentrations of PIC and BSi represent well the (historical) production of biogenic material from coccolithophores and diatoms, as well as the other groups examined here, their relationship to 'living biomass' is not necessarily direct and may break down seasonally and spatially. Recognising that this may occur, for example in post-bloom conditions, is an important caveat that should be clear to the reader.

Agreed, and we added the following sentence to the Introduction: We also note that our technique does not distinguish between living and non-living biomass, and thus is more representative of the history of production than the extent of extant populations at the time of sampling.

Ln 107-108: The calcite content of these different strains of Emiliania huxleyi also differ significantly (see Poulton et al., 2011 for estimates or Muller et al., 2015 for measurements), which may have strong implications for PIC production in S Ocean coccolithophore blooms (e.g. Poulton et al., 2013). Agreed, and we have added a sentence acknowledging this issue and these results: Of course, Emiliania huxleyi itself comes in several strains even in the Southern Ocean, with differing physiology, including differing extents of calcification [Cubillos et al., 2007; M. N. Muller et al., 2015; M.N. Muller et al., 2017; Poulton et al., 2013; Poulton et al., 2011].

Ln 263: Missing full stop between 'cell' and 'Calibration'. Full stop inserted.

Ln 468-470: The POC:PIC ratio given is relatively low, especially for the S Ocean strain: Muller et al. (2015) reports values of 0.83 for over-calcified strains, 1.5 for normal Atype and greater than 2 for the B/C type reported in the S. Ocean. Maybe the authors could add in a statement on the sensitivity of their estimates to cell POC:PIC ratios– and also how detached coccoliths may actually counteract high cellular POC:PIC ratios.A

Agreed. We have replotted Figure 4b using the POC:PIC ratio of 1.5 for the Southern Ocean morphotype A, and added discussion on latitudinal variations in morphotypes,

associated POC:PIC ratios, and their implications for our conclusions: The relatively small POC contribution from coccolithophores is only weakly sensitive to the ~3-fold variation [M. N. Muller et al., 2015] of POC/PIC ratios among Emiliania huxleyi morphotypes. Using the lower value of 0.83 observed for over-calcified forms that occur in the northern SAZ would reduce the POC contribution there but still leave it co-dominant with diatoms, and using the higher value of 2.5 observed for polar morphotype C would increase the POC contribution in Antarctic waters, but still leave it overwhelmed by the diatom contribution (Figure 4b). The relative contributions to total POC are also sensitive to the POC/PIC ratio chosen for diatoms (which vary significantly across genera; [O. Ragueneau et al., 2002; Olivier Ragueneau et al., 2006]). For these reasons, the relative dominance is best viewed on the log scale of Figure 4, and while keeping in mind the considerable scatter.

Ln 518-519: Biometric measurements have confirmed the low PIC per coccolith for the different morphotypes/strains (see Poulton et al., 2011 and/or Charalampopoulou et al., 2016; see also Muller et al., 2015 (as cited)). Agreed, and we have added a clause acknowledging these results: Early work in the South Atlantic found that SPIC values appeared to exceed ocean PIC by a factor of 2-3 [W M Balch et al., 2011], and based on a handful of samples it was suggested that this might reflect a lower amount of PIC per coccolith [Holligan et al., 2010], and it has since been confirmed that polar coccolithophores can have low PIC contents [Charalampopoulou et al., 2016; M. N. Muller et al., 2015; Poulton et al., 2011].

Ln 573: Please correct Emiliania Huxleyi to Emiliania huxleyi. Corrected as requested

Ln 622: Light utilization may be another important factor as there are pigment differences between E. huxleyi strains (see Cook et al., 2011) We added this possibility to the existing sentence, citing the work of Zhang et al., 2015, who measured light responses for coccolithophores: Coccolithophores, especially the most common species Emiliania huxleyi, have been studied sufficiently in the laboratory to allow possible important controls on their niches and especially their calcification rates to be proposed,

including temperature, pH, pCO2, calcite saturation state, light, and macro- and micro-nutrient availability [Bach et al., 2015; Feng et al., 2016; Mackinder et al., 2010; M. N. Muller et al., 2015; Müller et al., 2017; Schlüter et al., 2014; Schulz et al., 2007; Sett et al., 2014; Zhang et al., 2015].

Ln 653-655: Charalampopoulou et al. (2016) concluded that temperature and light were strong drivers of coccolithophore distribution and calcification across a latitudinal transect in Drake Passage (whilst also acknowledging the role of iron). Have the authors considered the role of (seasonal) light availability? We added this possibility and citation, while also stating that we did not have data sufficient to consider it further: Many properties that might influence coccolithophore productivity decreased strongly and close to monotonically from north to south across the Southern Ocean for our voyages (Figure 6). These include temperature (from 23 to -0.4 C for our samples), salinity (from 35.6 to 33.6, with tight correlation with alkalinity, not shown - data available in the Supplementary Material), pH (from 8.20 to 8.08 on the free scale), and the saturation state of calcite (from 5.22 to 2.12). The strong correlation of these properties means that it is not easy to separate their possible influences on coccolithophore distributions, without relying on specific thresholds or quantitative response models. This problem of correlations among drivers has been noted before in examining transect data across Drake passage, where more detailed measurements of coccolithophore properties augmented with incubation studies found temperature and light were the most probable drivers of coccolithophore abundance and calcification rates [Charalampopoulou et al., 2016]. Our lack of information on the availability of light (mixed layer depth was determined only on the two hydrographic sections), iron, or individual species and strains, makes deducing a possible influence of ocean acidification on coccolithophore distributions from our spatial distribution data even more difficult. We also reiterated the probable importance of light at the end of this section: Further progress in understanding the controls on coccolithophore abundances in the Southern Ocean is clearly needed. At present temperature, light, and competition with diatoms for iron appear to be the strongest candidates (at least for southward expansion [Charalampopoulou et al., 2016; Gafar et al., 2017]; with nitrate a strong influence on the location of the northern oligotrophic boundary; [Feng et al., 2016]).

---

## Author Response (AR2)

PICPOCBSI Paper revisions

1. Uploaded new version of data table "picpocbsi_5voyage_finaldata_ver003.txt". New table includes volumes of water filtered and filtrations times to help respond to question about filter blockage. Going back through the data discovered underreporting of flowmeter counts for Totten voyage samples. Recalculated volumes. In all cases this reduced PIC concentrations slightly which will not affect conclusions, in fact it will strengthen them. Also discovered two errors in the interpretation of SIPEXII handwritten logsheets which lead to unlikely flow rates being recorded for PIC. Samples were discarded from final dataset. R code now pulls data directly from the database rather than voyage specific tabs. There is now only one source of data which is easier to keep up to date and data extraction code is more consistent across voyages. In the event that the database is modified (hopefully not again) re-extraction of the data is easy. During the course of these realised Excel was rounding numbers off to what was displayed on screen when exporting as .txt file. Increased digits to 15 for all data columns for export to .txt so numbers as slightly different to first "final dataset" but will be more accurate. In final data set after calculations completed rounded lat longs, sal and temp to 4 digits, data columns to 8 digits, water volumes and filtrations times to 1 digit.

2. Table 1
   a. added superscript three to other column titles as suggested in comments.
   b. Updated number of samples to fit new version of data where a few samples were removed for QC reasons when flow rates and logsheets were re-examined.

3. Fig 1 Replotted with new better QCed dataset. Changed incorrect legend title from "Fronts" to "Voyage Leg".

4. Fig 3 Replotted with new better QCed dataset

5. Fig 4 replotted with new better QCed dataset

6. Fig 5 Replotted with new better QCed dataset. Rewrote code so panelling is done within R rather than manually.

7. Fig 6 did not need replotting because VL3 and VL6 data were not changed

8. Fig 7. Fixed minor issue with presentation of NOBM data. Changed plotting from geom_raster to geom_tile. Suspect lat longs in data not completely regular. geom_tile slower but makes sure tile is properly centred around coordinates.

9. New figure added, comparison of satellite SPIC and ocean PIC. Figure numbers adjusted. Discussion altered accordingly.

10. Fig S1 Replotted with new better QCed dataset. Rewrote code so panelling is done within R rather than manually. In original submission suspect VL8 satellite-PIC was incorrectly substituted for satellite-Chla during manual panelling (R code looks OK). This has now been fixed. Units are now more consistent PIC umol/L and Chl-a ug/L. Reduced range of Chla from 0-2ug/L to 0-1ug/L. This gives better colour in the map and not losing much at the top end.

11. All figures changed to colour-blind palette

11. corrections to the methods and reduced duplication in Smit BSi.

12. Technical Corrections requested by Kai Schulz:

   1.Introduction: low alkalinity changed to moderate alkalinity

   2.  POC/PIC changed to POC/BSI….thankyou!!

   3. Gafar et al (2017) amended.

Response to Reviewer 1: L. Bach Comments

Review on: "Distribution of planktonic biogenic carbonate organisms in the Southern Ocean south of Australia: a baseline for ocean acidification impact assessment " by Trull et al. In this study, Trull et al., investigate Diatom and calcifier distribution patterns in the Southern Ocean. Their analysis is based on BSi, POC and size fractionated PIC data. They compare their ground truth data with satellite data and model predictions and report important discrepancies and consistencies. I think their study is very valuable and their paper contains key information to document climate change effects on diC1 BGD Interactive comment Printer-friendly version Discussion paper atoms and calcifiers in the Southern Ocean. I really only have minor comments. Some of these are addressing their methods and some refer to the discussion/conclusion part.

**Line 26: Are diatoms really the most abundant phytoplankton? I can understand that they might be dominant in terms of biomass but would intuitively assume that smaller groups (e.g. picoeukaryotes such as Micromonas) are more abundant than diatoms. (I may be wrong here but just to double check.)**

Reviewer is correct. Sentence modified as follows:

*Ancillary measurements of biogenic silica (BSi) and particulate organic carbon (POC) provided context, as estimates of the biomass of diatoms (the highest biomass phytoplankton in polar waters), and total microbial biomass, respectively.*

**Line 56: I am not sure that the under-saturation is primarily due to low TA. I would assume that it is due to the low temperature that leads to generally low carbonate ion concentration.**

Reviewer is correct – the dominant effect is temperature.  For example, (based on CO2SYS with standard default constants), for seawater Salinity=35, Alkalinity=2320 umol/kg waters in equilibrium with pCO2=400 uatm air, cooling from 15 to 5 C reduces the carbonate anion concentration from 160 to 112 umol/kg, whereas at 15C dilution of salinity from 35 to 33 and alkalinity proportionally reduces carbonate anion concentrations from 160 to 145 umol/kg.  That is, temperature accounts for ~80% of the total effect.  Sentence modified as follows:

*The low temperature and low alkalinity of Southern Ocean waters make this region particularly susceptible to ocean acidification, ….*

**Line 63: It is a bit weird that you say that their relative importance is poorly known but then in the same sentence say that they will have an influence ecosystem health. The second part of the sentence implicitly contradicts the first part. Furthermore, I did not understand how the "importance" will "influence of the overall impact". This sentence could perhaps be rephrased.**

Agreed, and we have reordered and rewritten these sentences to make the issue clearer:

*Carbonate forming organisms in the Southern Ocean include  coccolithophores (the dominant carbonate forming phytoplankton; e.g. [Rost and Riebesell, 2004]), foraminifera (the dominant carbonate forming zooplankton; e.g. [Moy et al., 2009; Schiebel, 2002]), and pteropods (a larger carbonate forming zooplankton, which can be an important component of fish diets; e.g. [Doubleday and Hopcroft, 2015; Roberts et al., 2014]).  However, the importance of carbonate forming organisms relative to other taxa is unclear in the Southern Ocean [Watson W. Gregg and Casey, 2007b; Holligan et al., 2010].*

Trull, Tom (O&A, H…, 5/10/2017 3:10 PM

Trull, Tom (O&A, H…, 5/10/2017 3:10 PM

**Line 80: Aren't these results? Perhaps move this sentence to results part. Furthermore, I do not understand the use of the second "suggested" in this sentence. Please check.**

Agreed. Result sentence removed.

**Line 91: In this context it may also be useful to remind the reader that the PIC50 fraction could also contain aggregated coccolithophore calcite (e.g. within fecal pellets).**

We don't think this is likely, and accordingly we have not added this possibility to the text. The PIC50 fraction is collected at a high flow rate, sufficient to disaggregate most faecal matter. This perspective was corroborated by not seeing faecal pellets during visual inspection of the 50um mesh screens to remove rare zooplankton. We added text explaining this perspective in the Methods section:

*Based on visual examination, the high flow rate through the 50 μm nylon mesh was sufficient to disaggregate faecal pellets and detrital aggregates.*

**Line 152: It would be helpful to know whether or not you expect a loss of CaCO3 by sieving the samples. Are there large quantities of CaCO3 expected in the >1000 μm size fraction?**

In our experience the >1000μm fraction rarely contains anything, and early attempts to analyse this fraction yielded negligible CaCO3. The filter is occasionally useful for preventing krill and other large zooplankton from entering the filtration system. We have not tried to assess what would be "expected" in this large fraction, because we are not aware of data that would make this possible and we consider that the ship intake is unlikely to provide an unbiased sampling of organisms of this size which are often both mobile and rare. We made no changes to the text.

**Line 152: What do you mean by "ship clean"? Please clarify.**

Text has been added to clarify the meaning:

*All samples were collected from the ships' underway "clean" seawater supply lines with intakes at ~4 m depth. These supply lines are separate from the engine intakes, have scheduled maintenance and cleaning, and are only turned on offshore (to avoid possible contamination from coastal waters).*

**Line 154: Can you provide any information if the 50 μm filter tended to block when such a large volume is filtered? I am asking because it could be that towards the end of the filtration process also smaller particles might have been retained on the filter due to clogging. I know this is difficult to reconstruct, but in case you have any further information it would be useful to provide them. I have personally made bad experiences with sequential filtrations.**

The short answers are that:

1. We were aware of this potential problem and designed our filtration processes to minimize it. We do not consider that clogging was a problem.
2. If clogging retained PIC on the 50 um filter, than our PIC01 estimates would too low and our PIC50 estimates would be too high. Because the fraction of total PIC on the PIC50 mesh was generally quite small (10% or less), this possible redistribution does not affect any of our conclusions.

The longer answer is that it is challenging to create a filtration system capable of filtering water across the diverse conditions of the entire Southern Ocean. Our filtration system evolved over time, partly to deal with the issue of filter clogging. The first leg (VL1) was a purely sequential filtration system where the volume of water filtered was sometimes limited by either filter clogging, so that insufficient material was obtained on the other filter. For this reason prior to VL2 we added a pressure relief valve between the 50um and 1um filters which allowed large volumes of water to pass through the 50um filter and bypass the 1um filter. The second improvement prior to VL6 was the introduction of digital flow meters which recorded instantaneous flow rates. The final improvement prior to VL8 was the introduction of electronically controlled ball valves that stopped filtration when flow rates fell below threshold values (0.5L/min hiflo and 0.05L/min loflo). We believe these stop thresholds are very conservative and the filters are not truly clogged at this stage.

Using our final configuration of the system, on VL8 & VL9, samples from the lower latitudes (approx. 44S - 50S) have reasonably high levels of 1-50μm particles which can reduce flow rates through the 1 μm filter below our cut-off threshold of 0.05L/min prior to the 2 hour filtration time limit (Figure 1). However, very large volumes of water pass through the 50 μm filter (Figure 1). In mid latitudes (approx. 50S – 58S) flow rates remain high through both filters and most samples filter for 2 hours (Figure 1). At high latitudes (approx. 58S - 68S) flow rates through the 50 μm filter often reduced rapidly to the 0.5L/min cut-off threshold because the 50μm filter collects many large chain forming diatoms. In our filtration system this shuts down filtration through both filters which is reflected in the low filtration times and volumes (Figure 1). Ideally we would like to filter larger volumes of water at high latitudes which may require a filtration system capable of switching to a second 50 μm filter or something similar.

**Figure 1**

[Figure]

[Figure]

For VL8 & VL9 with their conservative cut-off thresholds we believe that clogging of the 50 μm filter and retention of smaller particles is unlikely. The picture is less clear for earlier legs where instantaneous flow rate data was not available and flowrate cut-off thresholds were not used. However, the data across all voyages shows quite consistent trends in PIC concentration and the PIC50/PIC01 ratio.

We have added these filtration times and volumes for all PIC samples in Table S1, included this discussion and figures in the Supplementary Materioal, and added a sentence into the main text pointing to this material:

*The flow rate and flow volume data also suggests that filter clogging was uncommon (see the Supplementary Information for expanded discussion).*

**Line 158: I am a bit nervous about the PIC filter cleaning procedure. Omega is 0 in the deionized water and the pH is (probably) low. Does the deionized water have the potential to dissolve CaCO3?**

We also were nervous about this. Accordingly we used degassed de-ionized water (boiling to remove CO2 and obtain close to neutral pH). The contact time of seconds and no loss of sharp edges on coccolithophores collected in this way and examined by scanning electron microscopy (Cubillos et al., 2007) reassured us. We added the following text:

*We consider that this rinse did not dissolve PIC, based on the sharp (non-eroded) features of coccolithophores collected in this way and examined by scanning electron microscopy (Cubillos et al., 2007).*

**Line 336: "as resulting" twice.**

Deleted one "as resulting"

**Line 402: I do not understand why mesoscale variability makes the comparison difficult. If you are at a certain location with a ship and sample PIC and you have satellite data for the very same time, you could easily compare these values, couldn't you?**

Strong mesoscale variability means that the match-up length scale must be very small. This limits the amount of match-ups that can be achieved. The variability length scale can also be smaller than the satellite pixel size. The correlation length scale for chlorophyll in the Southern Ocean degrades at distances > 10-15 km, as recently shown in attempting to match Biogeochemical-Argo fluorescent chlorophyll and satellite ocean colour estimates for a large set of observations (**Haëntjens N, Boss E, Talley LD (2017) Revisiting Ocean Color algorithms for chlorophyll a and particulate organic carbon in the Southern Ocean using biogeochemical floats. Journal of Geophysical Research: Oceans 122:6583-6593).**

Using a somewhat longer match-up length scale of 25 km (i.e. the ship and satellite observations must be within 25 km of each other on the same day), we were able to retain 116 match-ups and we have added these results to the paper. The match-ups tend to occur in clusters of several samples along a transit when the ship encountered cloud-free conditions, so that the amount of independent observations is less than this. Nonetheless the match-up results are valuable and they confirm that the satellite SPIC values are reasonable estimates in Subantarctic waters but very much too high in Antarctic waters.

The new results are described by a new figure and new text:

*Both cloudy conditions and strong mesoscale variability limit the number of direct comparisons (match-ups) that can be made. Using a match-up length scale of 25 km (i.e. the ship and satellite observations must be within 25 km of each other on the same day), which is somewhat larger than the correlation length scale for chlorophyll in the Southern Ocean of 10-15 km [Haëntjens et al., 2017], allowed us to retain 116 match-ups. These results, shown in Figure 6, confirm that the satellite SPIC values are reasonable estimates in Subantarctic waters, within a factor of 2-3 [W M Balch et al., 2011], but very much too high in Antarctic waters.*

**Line 405: What is "e-folding"? The term has not been introduced.**

This is a common term to describe exponential behaviour, e.g. from Wikipedia:

"In science, e-folding is the time interval in which an exponentially growing quantity increases by a factor of e; it is the base-e analog of doubling time."

No change was made to the text.

**Line 469: Dominant in terms of abundance? Dominant in terms of biomass would probably be the more important metric here.**

Agreed, and we changed "abundance" to "biomass" here and throughout this paragraph.

**Lines 470 and 473 : These results imply that diatoms (and to a limited extent coccolithophores) more or less exclusively contribute to the bulk POC in Antarctic waters. I am not so experienced with the plankton communities in the Southern Ocean but would intuitively disagree. Is it really possible that diatoms are so dominant? What about grazers? Did the analysis include e.g. copepod as a POC source or were these not captured on the filters? I think the result of bulk POC = diatom POC in the Antarctic is very interesting.**

We have added a qualifying sentence as follows (*in italics here but not in text*):

As shown in Figure 4b, this suggests that diatoms dominate the accumulation of organic carbon throughout the Southern Ocean, with coccolithophores generally contributing less than half that of diatoms in the SAZ and less than a tenth of that in Antarctic waters. *This statement is of course limited to POC captured by our small volume, size limited (1-1000 um) sampling procedure, and variability in the extent of dominance and the scaling of POC to biogenic minerals still allows significant contributions from other POC sources.*

**Line 480: Abundance of calcifiers or concentration of CaCO3? I think you should stick to the latter term to be more precise.**

Agreed, and sentence changed to:

*Finally, we note that the relatively low levels of PIC across the Southern Ocean as observed here means that POC/PIC ratios are high, greater than 4 in the SAZ and ranging up to 20 in Antarctic waters (Figure 4a).*

**Line 482: You argue that PIC/POC is low which leads to little influence on the TA mediated reduction of atmospheric CO2 uptake. I agree with that. However, PIC can induce biogeochemical feedbacks in other ways e.g. through ballasting (as you mention in the paragraph before). So I think that it is not really valid to say that coccolithophores had a limited influence on the uptake capacity of atmospheric CO2 if you neglect other feedback mechanisms than TA reduction.**

We agree with the reviewer on the multiple mechanisms of influence of calcification on air-sea CO2 transfer, and modified this sentence to make clear that only the aspect of alkalinity affects of surface ocean pCO2 is under consideration:

*This suggests calcification has a negligible countering impact on the reduction of surface ocean $CO_2$ partial pressure by phytoplankton uptake, even smaller than the few to ~10% influence identified earlier from deep sediment trap compositions in HNLC [P. W. Boyd and Trull, 2007a] and iron-enriched waters, respectively [Salter et al., 2014].*

**Section 3.4: In section 3.4 you compare model predictions with field data to test whether they predict meaningful trends. I think this is extremely valuable. I have, however, two comments.**

1) **You first use the Bach et al., 2015 and Langdon et al. 2000 models. These models only consider carbonate chemistry conditions and no other environmental parameter to predict calcification rates. Your data nicely shows that carbonate chemistry is obviously not the driving factor behind the latitudinal trend in the Southern Ocean because model prediction and latitudinal patterns are inconsistent. The Bach et al., model basically predicts that the carbonate chemistry conditions are close to ideal throughout the Southern Ocean. The Langdon et al., model predicts a decline which reflects the trend in Omega. Both models describe calcification response to carbonate chemistry and not distribution patterns of calcifiers. The reason why I mention this is because at the end of this part of the paper you state: "Thus, and unsurprisingly, coccolithophore abundances are clearly not controlled by inorganic carbon chemistry alone" (Lines 603- 604). I could not agree more with this statement. However, the way this is formulated implies to some extent that your finding contradicts what we have concluded in our study. But this is not the case. In Bach et al. (2015) we wrote: "great care must be taken when correlating carbonate chemistry with coccolithophore dispersal because this is by no means the only parameter controlling it. Physical (e.g. temperature), other chemical (e.g. nutrient concentrations), or ecological (e.g. grazing pressure) factors will under many if not most circumstances outweigh the influence of carbonate chemistry conditions, unless differences in the latter are extreme. We will therefore focus the discussion on those cases where differences in carbonate chemistry conditions are rather extreme." Thus, our valuation is very similar to that of the authors of this manuscript. I would appreciate if you could point out that your main conclusion in this paragraph (that carbonate chemistry is probably not the key factor controlling coccolithophore distribution) is also in line with (and not conflicting with) what we assumed in our studies.**

2)

We are very happy to do this, and added this information explicitly in the summary of this section:

*Thus, and unsurprisingly, coccolithophore abundances are clearly not controlled by inorganic carbon chemistry alone. This perspective has been strongly emphasized previously, including by Bach et al., (2015), who noted " …great care must be taken when correlating carbonate chemistry with coccolithophore dispersal because this is by no means the only parameter controlling it. Physical (e.g. temperature), other chemical (e.g. nutrient concentrations), or ecological (e.g. grazing pressure) factors will under many if not most circumstances outweigh the influence of carbonate chemistry conditions…".*

**2) I am a bit skeptical about the growth rate vs. temperature argument based on the Norberg model. The model predicts a decline of coccolithophore growth rates due to decreasing temperature. This in itself is not convincing because the decrease of growth rate would apply for every other phytoplankton group as well. What you would really have to look at is if the growth rate of coccolithophores decreases over-proportionally relative to other phytoplankton species. If this was then case, then you could argue that coccolithophores become less competitive the further South you go.**

We made no changes in response to this comment, because while it has merit we were already careful to describe at the start of this paragraph that the presentation of the Norberg model was limited to an exploration of the possible response to temperature in a univariate sense:

"To provide a brief visualization of the expected univariate response, we fit the "Norberg" thermal optimum envelope model ….

and we already re-emphasize at the end of the paragraph that this exploration was limited in scope:

There are of course many other possible explanations (as noted at the start of this section).

In addition, in the following paragraphs we were already careful to note again that autotrophic completion was a larger issue - see our response to the next comment.

Accordingly, we made no further changes.

**Line 653: In this concluding remark you only consider the bottom-up control on diatom vs. coccolithophore distribution. Have you also considered if top-down mechanisms could have played a role here? Even though there may not be appropriate data available to test this in the present study, it may still be useful to remind the reader that this mechanism exists and could also have played a role. I think the Assmy et al., (2013) study nicely made the case that predators may have an important influence on phytoplankton composition in the Southern Ocean.**

We had already mentioned this possibility, but have augmented it with a final clause in parentheses to cite the Assmy et al., 2013 study:

Importantly, in addition to multivariate environmental control of coccolithophore distributions via their growth rates, there is the possibility of control by resource competition with other autotrophs (presumably mainly for iron) and/or stronger loss terms to grazers in Antarctic than Subantarctic waters ([*Assmy et al.*, 2013] has suggested preferential grazing as a control on community structure; but we have no data to allow us to evaluate this).

**Table 1: I think the uppercase 3 also needs to be added to PIC01, POC, and BSi.**

Agreed and we added upper case 3 to PIC01, POC, BSi

**Figure 1: It would be helpful to add full names and abbreviations of the various fronts to the figure caption.**

Full names and abbreviations added

**Figure 3: One particularly interesting finding presented in Figure 3 is that PIC50 (foraminifera) concentrations are considerably lower than PIC01 (coccolithophores) concentrations except for maybe the most Southern stretch of the transects. Sometimes the discrepancies are orders of magnitude. This implies that coccolithophores are the much more important pelagic calcifiers in the Southern Ocean than foraminifera and pteropods. Is this conclusion valid? If so, I think this finding definitely deserves more attention in your paper. Furthermore, it would be interesting to compare this with the results from Broecker**

**and Clark (2009) who found roughly equal contribution of coccolithophores and foraminifera to the sediment CaCO3 (although their most southern sample came from 40 South).**

We deliberately avoided discussion of the PIC50 distributions in any detail for multiple reasons, as we had stated early in the Introduction. Comparison to sediments would bring in the further complexity of the extent of losses of these organisms after leaving surface waters, and become very speculative. Accordingly, we have not added discussion on this issue, and instead have further strengthened our sentence regarding why we do not discuss these results in any detail in the revised version:

*We do not discuss the PIC50 results in any detail because of this complexity, because controls on foraminifera distributions appear to involve strongly differing biogeography of several co-dominant taxa, rather than dominance by a single species [Be and Tolderlund, 1971], because the numbers of these organisms collected by our procedures was small, and because assessing these issues is beyond the scope of this paper.*

**Figure 4: What was the rationale of showing the POC/PIC ratio? I think readers will generally be more familiar with PIC/POC ratios.**

Both are in common use.  We preferred POC/PIC (and BSi/PIC) because these emphasized our key findings that BSI and POC are both much more abundant that PIC.

We made no change.

Response to Reviewer 2: A. Poulton Comments

**My recommendation is that the authors need to further elucidate in the manuscript the distinction between particulate material and 'living biomass'. Most particulate pools (POC, PIC and BSi) contain variable contributions of both living biomass (i.e. organisms) and detrital material. In the case of PIC, detached coccoliths (e.g.) can be a significant fraction of the total pool and their small size conveys very slow sinking speeds. In the case of diatoms, empty or broken frustules may stay in suspension (or as part of the cell chain) after the organic material has been removed. Similar comments may be made about the larger biogenic organisms (foram fragments, juvenile shells, radiolarian tests). While the particulate concentrations of PIC and BSi represent well the (historical) production of biogenic material from coccolithophores and diatoms, as well as the other groups examined here, their relationship to 'living biomass' is not necessarily direct and may break down seasonally and spatially. Recognising that this may occur, for example in post-bloom conditions, is an important caveat that should be clear to the reader.**

Agreed, and we added the following sentence to the Introduction:

*We also note that our technique does not distinguish between living and non-living biomass, and thus is more representative of the history of production than the extent of extant populations at the time of sampling.*

**Ln 107-108: The calcite content of these different strains of Emiliania huxleyi also differ significantly (see Poulton et al., 2011 for estimates or Muller et al., 2015 for measurements), which may have strong implications for PIC production in S Ocean coccolithophore blooms (e.g. Poulton et al., 2013).**

Agreed, and we have added a sentence acknowledging this issue and these results:

*Of course, Emiliania huxleyi itself comes in several strains even in the Southern Ocean, with differing physiology, including differing extents of calcification [Cubillos et al., 2007; M. N. Muller et al., 2015; M.N. Muller et al., 2017; Poulton et al., 2013; Poulton et al., 2011].*

**Ln 263: Missing full stop between 'cell' and 'Calibration'.**

Full stop inserted.

**Ln 468-470: The POC:PIC ratio given is relatively low, especially for the S Ocean strain: Muller et al. (2015) reports values of 0.83 for over-calcified strains, 1.5 for normal Atype and greater than 2 for the B/C type reported in the S. Ocean. Maybe the authors could add in a statement on the sensitivity of their estimates to cell POC:PIC ratios– and also how detached coccoliths may actually counteract high cellular POC:PIC ratios.A**

Agreed. We have replotted Figure 4b using the POC:PIC ratio of 1.5 for the Southern Ocean morphotype A, and added discussion on latitudinal variations in morphotypes, associated POC:PIC ratios, and their implications for our conclusions:

*The relatively small POC contribution from coccolithophores is only weakly sensitive to the ~3-fold variation [M. N. Muller et al., 2015] of POC/PIC ratios among Emiliania huxleyi morphotypes. Using the lower value of 0.83 observed for over-calcified forms that occur in the northern SAZ would reduce the POC contribution there but still leave it co-dominant with diatoms, and using the higher value of 2.5 observed for polar morphotype C would increase the POC contribution in Antarctic waters, but still leave it overwhelmed by the diatom contribution (Figure 4b). The relative contributions to total POC are also sensitive to the POC/PIC ratio chosen for diatoms (which vary significantly across genera; [O. Ragueneau et al., 2002; Olivier Ragueneau et al., 2006]). For these reasons, the relative dominance is best viewed on the log scale of Figure 4, and while keeping in mind the considerable scatter.*

**Ln 518-519: Biometric measurements have confirmed the low PIC per coccolith for the different morphotypes/strains (see Poulton et al., 2011 and/or Charalampopoulou et al., 2016; see also Muller et al., 2015 (as cited)).**

Agreed, and we have added a clause acknowledging these results:

*Early work in the South Atlantic found that SPIC values appeared to exceed ocean PIC by a factor of 2-3 [W M Balch et al., 2011], and based on a handful of samples it was suggested that this might reflect a lower amount of PIC per coccolith [Holligan et al., 2010], and it has since been confirmed that polar coccolithophores can have low PIC contents [Charalampopoulou et al., 2016; M. N. Muller et al., 2015; Poulton et al., 2011].*

**Ln 573: Please correct Emiliania Huxleyi to Emiliania huxleyi.**

Corrected as requested

**Ln 622: Light utilization may be another important factor as there are pigment differences between E. huxleyi strains (see Cook et al., 2011)**

We added this possibility to the existing sentence, citing the work of Zhang et al., 2015, who measured light responses for coccolithophores:

*Coccolithophores, especially the most common species Emiliania huxleyi, have been studied sufficiently in the laboratory to allow possible important controls on their niches and especially their calcification rates to be proposed, including temperature, pH, $pCO_2$, calcite saturation state, light, and macro- and micro-nutrient availability [Bach et al., 2015; Feng et al., 2016; Mackinder et al., 2010; M. N. Muller et al., 2015; Müller et al., 2017; Schlüter et al., 2014; Schulz et al., 2007; Sett et al., 2014; Zhang et al., 2015].*

**Ln 653-655: Charalampopoulou et al. (2016) concluded that temperature and light were strong drivers of coccolithophore distribution and calcification across a latitudinal transect in Drake Passage (whilst also acknowledging the role of iron). Have the authors considered the role of (seasonal) light availability?**

We added this possibility and citation, while also stating that we did not have data sufficient to consider it further:

[revised manuscript text omitted]

Passmore, Abe (O…, 27/9/2017 9:11 AM

Passmore, Abe (O…, 27/9/2017 9:12 AM

Passmore, Abe (O…, 27/9/2017 9:13 AM

Trull, Tom (O&A, H…, 9/10/2017 2:39 PM
Trull, Tom (O&A, H…, 9/10/2017 2:40 PM
Passmore, Abe (O…, 27/9/2017 9:14 AM

Trull, Tom (O&A, H…, 9/10/2017 3:17 PM
**Moved (insertion) [2]**
Trull, Tom (O&A, H…, 9/10/2017 3:48 PM

there but still leave it co-dominant with diatoms, and using the higher value of 2.5 observed for polar morphotype C would increase the POC contribution in Antarctic waters, but still leave it overwhelmed by the diatom contribution (Figure 4b).  The relative contributions to total POC are also sensitive to the POC/BSi ratio chosen for diatoms (which vary significantly across genera; [*O.*

*Ragueneau et al.*, 2002; *Olivier Ragueneau et al.*, 2006]).  For these reasons, the relative dominance is best viewed on the log scale of Figure 4b, and while keeping in mind the considerable scatter.

Figure 4b also emphasizes that total POC contents can be largely explained by diatom biomass in

Antarctic waters (south of 50 °S), whereas in the SAZ (north of 50 °S), total POC often exceeds the sum of contributions from diatoms and coccolithophores.  This serves as an important reminder that other organisms are important to the carbon cycle in the SAZ, and phytoplankton functional type models should avoid over-emphasis on diatoms and coccolithophores just because they have discernable biogeochemical impacts (on silica and alkalinity, respectively) and satellite remote sensing signatures [*Hood et al.*, 2006; *Moore et al.*, 2002].  Finally, we note that the relatively low levels of PIC across the Southern Ocean as observed here means that POC/PIC ratios are high, greater than 4 in the SAZ and ranging up to 20 in Antarctic waters (Figure 4a).  This suggests calcification has a negligible countering impact on the reduction of surface ocean $CO_2$ partial pressure by phytoplankton uptake, even smaller than the few to ~10% influence identified earlier from deep sediment trap compositions in HNLC [*P. W. Boyd and Trull*, 2007a] and iron-enriched waters, respectively [*Salter et al.*, 2014].

Notably, our Southern Ocean PIC01 estimates are smaller than those found in northern hemisphere polar waters.  As compiled by Balch et al. (2005), concentrations were 100-fold higher (~10 μM) in the north Atlantic south of Iceland (60-63 °N) than any of our values, and 1000-fold higher than our values in the same southern hemisphere latitude range.  Values collected over many years from the

Gulf of Maine [*W M Balch et al.*, 2008] were ~ 1 μM, and thus 5-10 times higher than our SAZ values (Gulf of Maine summer temperatures are similar to the SAZ, and colder in winter).  This difference between hemispheres is also evident in observations from the South Atlantic, where PIC values estimated from acid labile backscatter for 6 voyages between 2004 and 2008 and latitudes 40-50 °S

were ~0.1-0.5 μM in remote waters [*W M Balch and Utgoff*, 2009], increasing to 1-2 μM in the

Argentine Basin with a few values reaching 4 μM [*W Balch et al.*, 2014].  These high South Atlantic observations are the highest of the "Great Calcite Belt" identified as a circumpolar feature of

Subantarctic waters based on SPIC values [*W Balch et al.*, 2014; *W M Balch et al.*, 2011].  Notably, shipboard PIC measurements in this feature are 2-3 times lower than the SPIC estimates in the South

Atlantic [*W M Balch et al.*, 2011], and ship collected samples from two voyages across the South

Atlantic and Indian sectors [*W M Balch et al.*, 2016] exhibit PIC concentrations (actual PIC values

Trull, Tom (O&A, H…, 9/10/2017 3:17 PM
**Moved up [2]:** [*M. N. Muller et al.*, 2015]

Passmore, Abe (O…, 27/9/2017 9:14 AM

Trull, Tom (O&A, H…, 5/10/2017 9:45 PM

Passmore, Abe (O…, 27/9/2017 9:42 AM

Passmore, Abe (O…, 27/9/2017 9:16 AM

Trull, Tom (O&A, H…, 5/10/2017 9:45 PM

Trull, Tom (O&A, …, 5/10/2017 10:02 PM

Trull, Tom (O&A, …, 5/10/2017 10:12 PM

Passmore, Abe (O…, 27/9/2017 9:17 AM

Passmore, Abe (O…, 27/9/2017 9:18 AM

Trull, Tom (O&A, …, 5/10/2017 10:05 PM

Passmore, Abe (O…, 27/9/2017 9:19 AM

Trull, Tom (O&A, …, 5/10/2017 10:05 PM

Passmore, Abe (O…, 27/9/2017 9:19 AM

Passmore, Abe (O…, 27/9/2017 9:19 AM

Passmore, Abe (O…, 27/9/2017 9:19 AM

Passmore, Abe (O…, 27/9/2017 9:20 AM

Passmore, Abe (O…, 27/9/2017 9:20 AM

Passmore, Abe (O…, 27/9/2017 9:20 AM

[revised manuscript text omitted]

Diana Davies 3/11/2017 4:38 PM

Passmore, Abe (O…, 27/9/2017 9:35 AM

Passmore, Abe (O…, 27/9/2017 9:35 AM

Passmore, Abe (O…, 27/9/2017 9:35 AM

Trull, Tom (O&A,…, 10/10/2017 11:51 AM

Diana Davies 3/11/2017 4:38 PM

Passmore, Abe (O…, 27/9/2017 9:35 AM

Passmore, Abe (O…, 27/9/2017 9:36 AM

Passmore, Abe (O…, 27/9/2017 9:36 AM

Trull, Tom (O&A,…, 10/10/2017 11:52 AM

Trull, Tom (O&A,…, 10/10/2017 11:52 AM

Trull, Tom (O&A,…, 10/10/2017 11:52 AM

Passmore, Abe (O…, 27/9/2017 9:37 AM

in the Scotia Sea [*Holligan et al.*, 2010], and in the South Atlantic and South Indian Oceans, especially in regions of natural iron fertilization [*W M Balch et al.*, 2016; *Smith et al.*, 2017]. In the NOBM, diatoms are also simulated and show (Figure 8) the expected high abundance in Antarctic waters in the southern third of the Southern Ocean, decreasing northward as in our results (but also show a band of elevated diatom concentrations in the Subantarctic, which we did not observe).

Competition for nutrients in the NOBM favours the ability of coccolithophores over diatoms to get by on limited resources (half-saturation constants for nitrate and iron of 0.5 and 0.67 versus 1.0 and 1.0 $\mu M$) including light (half saturation constant of 56 versus 90 $\mu$mol photons m$^{-2}$ s$^{-1}$ under Southern Ocean low light conditions). But diatoms are specified to have higher growth rates when all resources are non-limiting (maximum growth rate at 20 °C 1.50 versus 1.13, both with the same Eppley dependence on temperature). Thus in the model, diatoms dominate silicon replete Southern Ocean waters, outcompeting other species for the limiting iron, and only give way to other species when silicon is depleted. Notably these other species then do best when additional Fe is supplied from either atmospheric sources (in the north where continental dusts are not shielded by ice) or island oases such as Crozet or Kerguelen. This view is compatible with our observations and those carried out in the northern half of the Southern Ocean during the "Great Calcite Belt" voyages [*W M Balch et al.*, 2016; *Smith et al.*, 2017]. It suggests that potential expansion of coccolithophores southward might be linked to decreasing supply of silicon from reduced upwelling of Circumpolar Deep Water in a progressively more stratified global ocean. A cautionary note to this conclusion is provided by the NOBM simulation of significant concentrations of diatoms in the SAZ where silicon is low, which arises from their specified higher maximum growth rate, emphasizing the importance of this parameter, and its temperature dependence, in modeling phytoplankton distributions. In specifying this temperature dependence, this model and most others still rely on the global compilation from nearly 50 years ago [*Eppley*, 1972]. Clearly better understanding of the controls on maximum growth rates and their temperature tolerance for key phytoplankton taxa is needed, first to understand current distributions and then to explore possible future changes.

**4. Conclusions**

Our surveys of PIC concentrations as a proxy for coccolithophores in the Southern Ocean south of Australia suggest:

- The concentrations of coccolithophores were much smaller (at least 10-fold) in the open Southern Ocean south of Australia than in northern hemisphere oceans.

- Coccolithophores were most abundant in the SAZ, and occasionally in the PFZ.

- The contribution of coccolithophores to total phytoplankton biomass (estimated from POC) was small, less than 10% in Subantarctic waters and less than 1% in Antarctic waters.

- The "Great Calcite Belt" characterization of SAZ and PFZ waters is overstated south of Australia, because both the satellite (SPIC) estimates and our in-situ PIC measurements show lower values than in the South Atlantic and South Indian where this feature was first suggested.

- The satellite PIC (SPIC) algorithm provides a good estimate, within a factor of 2-3, of PIC values in Subantarctic waters south of Australia, but erroneously suggests large agglomerations of PIC in polar waters, where little to none is present south of Australia.

- Our PIC results and ancillary measurements of biogenic silica, particulate organic carbon, dissolved nutrients, and inorganic carbon system status may be useful in the testing of models of limiting conditions and ecological competitions that affect coccolithophore distributions. Preliminary considerations suggest that temperature, iron, and competition with diatoms may be stronger influences than pH or calcite saturation state.

Despite the considerable effort required to obtain these survey results, much remains to be done just to define coccolithophore distributions, for example their seasonality, especially when the complexities of differing responses of individual species and strains are considered.

Trull, Tom (O&A, H…, 9/10/2017 5:45 PM

Trull, Tom (O&A, H…, 9/10/2017 5:46 PM

Trull, Tom (O&A, H…, 9/10/2017 5:46 PM

Trull, Tom (O&A, …, 5/10/2017 10:29 PM

Trull, Tom (O&A, H…, 9/10/2017 5:43 PM

Trull, Tom (O&A, H…, 9/10/2017 5:43 PM

Trull, Tom (O&A, H…, 9/10/2017 5:44 PM

Trull, Tom (O&A, H…, 9/10/2017 5:44 PM

Passmore, Abe (O…, 27/9/2017 9:52 AM

Passmore, Abe (O…, 27/9/2017 9:52 AM

Passmore, Abe (O…, 27/9/2017 9:53 AM

[revised manuscript text omitted]